

**Measurement report: Characteristics and sources of non-methane**
**VOCs and their roles in SOA formation during autumn in a central**
**Chinese city**
Haixu Zhang[1,2], Chunrong Chen[1], Weijia Yan[1], Nana Wu[1], Yu Bo[3], Qiang Zhang[1], and
Kebin He[2,4]
[1] Ministry of Education Key Laboratory for Earth System Modelling, Department of Earth System
Science, Tsinghua University, Beijing 100084, China
[2] State Key Joint Laboratory of Environment Simulation and Pollution Control, School of
Environment, Tsinghua University, Beijing 100084, China
[3] RCE-TEA, Institute of Atmospheric Physics, Chinese Academy of Science, Beijing 100029, China
[4] State Environmental Protection Key Laboratory of Sources and Control of Air Pollution Complex,
Tsinghua University, Beijing 100084, China
Correspondence: Haixu Zhang (zhanghaixu@tsinghua.edu.cn) and Yu Bo (boyu@mail.iap.ac.cn)



**Abstract**

Volatile organic compounds (VOCs) are essential in secondary organic aerosol
(SOA) formation due to their dual roles as precursors and oxidant producers. In this
work, the VOC species in Xinxiang, a mid-sized city located in Henan Province in
central China, were measured and analysed from November 5th to December 3rd, 2018.
Based on online monitoring with proton transfer reaction-mass spectrometry (PTR-MS)
and canister grab samples, 53 VOC species are obviously detected, and the most
abundant categories are oxygenated VOCs (OVOCs) and benzenoids. Compared with
field measurements in other regions, the mixing ratios of BTEX (benzene, toluene,
ethylbenzene, and xylene), acetaldehyde, and C3 carbonyls are at high levels, indicating
intensive anthropogenic emissions in Xinxiang. According to the positive matrix
factorization (PMF) model, benzenoids are mainly emitted from solvent evaporation
(~47%), residential heating (~19%), industrial emission (~16%), and vehicle exhaust
(~10%), while the contributions from biogenic and secondary sources as well as thermal
power generation are minor. However, the emissions of total OVOCs from the six
resolved sources are similar. The potential source contribution function (PSCF) and
concentration weighted trajectory (CWT) results show that the transport contribution
for VOCs is not intensive, but the cities within Henan Province or in the neighbouring
provinces may influence the mixing ratios to some extent. The roles of benzenoids and
OVOCs in SOA formation are investigated by estimating the mass of oxidation
products and rates of OH radical production. Among the observed VOCs, toluene has
the largest SOA formation potential (SOAFP), while its weight in SOA formation
declines with the aggravation of pollution. On the other hand, the SOA concentration
shows a good relationship with OH exposure, which highlights the importance of the
atmospheric oxidation capacity, especially in polluted periods. Formaldehyde is the
strongest radical contributor, and the contribution of acetaldehyde is also significant in
this study. Furthermore, solvent evaporation, industrial emissions, and vehicle exhaust
are estimated as the top three anthropogenic contributors with the highest SOAFP and
radical contribution rate.



## 1. Introduction


The haze problem in China has attracted much attention in recent decades. Many
observation studies have shown that secondary organic aerosol (SOA) comprises a
major fraction of the fine particle mass, and some recent studies highlight the driving
force of SOA in extremely severe pollution episodes (Crounse et al., 2013; Guo et al.,
2014; Huang et al., 2014). The level of SOA is primarily determined by both the
concentrations of gaseous precursors and atmospheric oxidizing capacity (Rao et al.,
2016). Therefore, volatile organic compounds (VOCs) are critical in SOA formation
because of their roles as precursors and active participants in the cycling of free radicals
(Atkinson et al., 2008; Kroll and Seinfeld, 2008; Lelieveld et al., 2008). Among
hundreds of VOC species, aromatic hydrocarbons constitute an important fraction
(~20–30%) of the urban atmosphere and have been suggested to be important SOA
precursors in many studies (Calvert, 2002; Ding et al., 2014; Yuan et al., 2013; Wu and
Xie, 2018). Oxygenated volatile organic compounds (OVOCs) are also an important
category of species because they are not only essential members of oxidation processes
but also the most important radical sources in polluted urban environments (Shao et al.,
2011; Kristensson et al., 2004; Emmerson et al., 2005; Edwards et al., 2014).
To aid in SOA estimation, many laboratory studies have investigated the SOA
yield of individual precursors. Accordingly, the ambient SOA yield cannot be
represented by a unique value because it is dependent on the organic aerosol mass
concentration, NOx concentration and temperature (Odum et al., 1996; Rollins et al.,
2012; Sarrafzadeh et al., 2016). The two-product model considering the above
influencing factors is a widely used SOA yield model in three-dimensional chemical
transport models (Appel et al., 2008; Tsimpidi et al., 2011; Li et al., 2015), but it is
scarcely applied to SOA estimation in field studies to our knowledge. Regarding
photooxidants, numerous measurements have been conducted focusing on the
performance of OVOCs at different stages of pollution and their potential sources
(Duan et al., 2012; Yang et al., 2017; Li et al., 2010; Liu et al., 2012), and only a few
studies have conducted quantitative analyses on radical production rates based on field





observations in China by utilizing a tropospheric ultraviolet and visible (TUV, version
5.0; http://cprm.acd.ucar.edu/Models/TUV/) radiation model (Rao et al., 2016; Wang
et al., 2017).
Vehicle exhaust emissions, industry emissions, fossil fuel volatilization, the use of
chemical reagents, and biomass combustion are important sources of atmospheric
VOCs, but there are some differences in different regions (Qi et al., 2014; Yang et al.,
2013; Li et al., 2019a; Liu et al., 2017; Zheng et al., 2018). In addition, the source
apportionment of OVOC sources, especially carbonyls, still has many uncertainties due
to the complex sources and sinks (Huang et al., 2019; Chen et al., 2014). The sources
show different chemical reactivities because of the VOC compositions. Hence, an
inventory-based SOA formation potential (SOAFP) list in China has been made to
identify the major species and sources contributing to SOA (Wu and Xie, 2018). In
some field studies, the SOAFP calculation is also applied to reveal the atmospheric
characteristics and the critical components for SOA increase, but few studies compare
the SOAFP among the resolved sources (Zhang et al., 2017; Zhang et al., 2018; Han et
al., 2020).
Xinxiang, a central city in China, has been within the most polluted region in
recent years. During the autumn of 2018, we conducted a field study focusing on the
characteristics of VOCs at an urban site in Xinxiang for the first time. In this work, the
mixing ratios, temporal variations, and diurnal patterns of the VOCs are shown and
compared with those at other sites. Source apportionment and regional contributions
are investigated based on the PMF model, as well as PSCF and CWT analyses. The
roles of benzenoids and OVOCs in SOA formation are investigated by estimating the
mass of oxidation products and rates of OH radical production, respectively, with
parameters (such as SOA yields and photolysis rates) based on real-time data. Finally,
the information about SOAFP as well as the radical producing capacity are first
assigned to the resolved source.
**2.  Experiment**
**2.1  Sampling site and measurements**



The measurements were performed in a mobile laboratory located in a square of
the Party School in Xinxiang. The sampling site was tens of metres away from a
national air quality monitoring site (35.3N, 113.9E) in the urban district (Hongqi
District). The surroundings were residential areas and colleges, except for a few
pharmaceutical factories 2 km to the west.
The VOC concentrations were observed by utilizing quadrupole proton transfer
reaction-mass spectrometry (PTR-QMS 500, Ionicon Analytik, Austria). The
observation principle and the deployment of PTR-MS have been described in many
previous studies (Lindinger et al., 1998; Yuan et al., 2013; Li et al., 2019b). Briefly,
only the species with a proton affinity greater than that of $H_2O$ (691 kJ mol$^{-1}$) can be
detected. In this work, the PTR-MS was operated at a standard condition: the pressure
of the drift tube was held at 2.2 mbar, and the temperatures of the inlet line and the drift
tube were both kept at 60 °C, with the reduced electric field parameter (E/N, where E
is the electric field and N is the gas number density) maintained at 135 Td. Air samples
were drawn through a Teflon line with an inner diameter of 0.125 cm. The VOC
measurements were performed in full-scan mode, browsing a large range of masses
(m/z 21.0—200.0), with a time resolution of ~10 s. In all, 44 mass peaks were involved
in this study, and the attribution of each peak to specific VOC species is summarized
in Table S1 of the supplement. The VOCs analysed as focus or tracers are 1. m/z 31
(formaldehyde); 2. m/z 33 (methanol); 3. m/z 42 (acetonitrile); 4. m/z 45 (acetaldehyde);
5. m/z 47 (formic acid and ethanol); 6. m/z 59 (acetone); 7. m/z 69 (isoprene); 8. m/z
(methyl vinyl ketone and methacrolein, MVK+MACR); 9. m/z 79 (benzene); 10.
m/z 93 (toluene); 11. m/z 105 (styrene); and 12. m/z 107 (C8 aromatics, including
ethylbenzene and xylenes). The PTR-MS was calibrated regularly with a dynamic
calibrator (Model 146i, Thermo Scientific, USA) using two standard gas cylinders
containing formaldehyde with a mixing ratio of 10 ppm and 17 other VOCs with a
mixing ratio of 1 ppm of each species. The instrument background calibration was
performed by installing a charcoal cartridge (Supelco, USA) upstream of the PTR-MS
inlet. PTR-MS Viewer software (Version 3.1, Ionicon Analytik, Austria) was used to



calibrate the transmission curve and process the observed data. More information on
the calibration process and the list of the 17 standard gases are shown in Text S1 of the
supplement.
In addition, 27 grab samples were collected using 3.2 L SUMMA canisters (Entech
Instrument, USA) with a sample duration of 1 h in the daytime, and the specific
sampling time is listed in Table S2. Before sampling, the canisters were precleaned with
high-purity nitrogen and pressurized to 50 psi, and one of them was then filled with
high-purity nitrogen as a blank sample. The chemical analysis was accomplished within
2 weeks after sampling, based on Compendium Method TO-15 (EPA, 1999). Briefly,
in this work, the air samples in the canisters were initially concentrated at -160 °C using
liquid nitrogen in a cryogenic preconcentrator (7100A, Entech, USA) to remove $CO_2$
and $H_2O$. Then, the samples were thermally desorbed at 120 °C and transported into a
GC-MS system (Model 7890A-5975C, Agilent Technologies, USA) with a DB-624
column (60 m×25 mm inner diameter with 1.4 μm film thickness) for analysis. The
standard gas of PAMS and TO 15 (1 ppm; Spectra Gases, USA) was used to construct
the calibration curves. In total, VOC species were effectively observed including 12
alkanes, 12 halohydrocarbons, and 14 aromatic hydrocarbons. No alkenes, alkynes or
C1-C2 alkanes were detected under the existing conditions.
The mass concentration of organic aerosol (OA) of NR-PM$_1$ was measured by an
Aerosol Chemical Speciation Monitor (ACSM, Aerodyne, USA), with a time resolution
of 15 min. The details of instrument operation and data analysis have been described in
previous studies (Ng et al., 2011; Li et al., 2018). The oxygenated OA (OOA) was
determined by utilizing the PMF model (Li et al., 2018), and the resolved results are
shown in Text S2 and Figs. S1-2. In this study, OOA is approximately treated as SOA.
The $SO_2$, CO, $NO_x$, and $O_3$ concentrations were measured by corresponding gas
analysers (Model 43i-TLE, 48i-TLE, 17i, and 49i, Thermo Scientific, USA). The PM$_{2.5}$
mass concentration was measured online by utilizing a heated tapered elemental
oscillating microbalance (TEOM series 1405, Thermo Scientific, USA), and the
meteorological conditions, including temperature, RH, pressure, wind speed, and wind



direction, were continuously reported by a portable weather station (WXT536, Vaisala,
Finland). In addition, the heights of the planetary boundary layer (PBL) were provided
by the local environmental monitoring station from their lidar station.
**2.2  Source apportionment**
The US EPA Positive Matrix Factorization (PMF) receptor model (version 5.0,
Sonoma Technology, Inc. USA), based on the multi-linear engine (ME-2) approach,
was used for source apportionment. The relevant parameters and calculation principles
have been described explicitly in previous studies (Norris G., 2014; Sarkar et al., 2017).
In the current work, 610 hourly averaged samples were involved, with 44 species
including 41 ion peaks from PTR-MS as well as CO, $SO_2$, and $NO_X$. The corresponding
uncertainties were calculated from the method detection limit (MDL) and the
determination error fraction, which is recommended by the user guide (Norris G., 2014).
Detailed information about the operation principle and the input data is described in
Text S3 and Table S1. The PMF was performed with 20 base runs, and the result with
the lowest Q (robust) value was chosen. To determine the optimum solution, PMF
model runs with 2 to 10 factor numbers were carried out.
**2.3  Trajectory analysis**
In this study, 48 h backward trajectory analysis with 1 h intervals (starting from
00:00 to 23:00 local time, LT) was performed each day at a height of 500 m above
ground level with the Hybrid Single-Particle Lagrangian Integrated Trajectory
(HYSPLIT) model (http://www.noaa.gov, last access: 10 February 2020 ). The results
were saved as endpoint files and further processed for trajectory clustering and statistics
using       MeteoInfo       software       plugged       in       with       TrajStat
(http://www.meteothinker.com/downloads/index.html, last access: 10 February 2020 ).
The spatial distributions of the identified sources were thereby analysed with PSCF and
CWT, which indicated the proportion of the source contribution in a given grid and the
concentration levels of the trajectories, respectively. In this study, the domain area was
in the range of 32-42ºN, 100-120ºE with a resolution of 0.5º×0.5º. More details on the
description can be obtained in Text S4.



**2.4 SOA production estimation**

In this study, a parametrization method was used to estimate the SOA formation potential (SOAFP) as well as the real-time SOA production, and the formula is listed as follows:

$$SOAFP_i = VOC_i \times Y_i \tag{1}$$

$$SOA_i = VOC_{i,consumed} \times Y_i \tag{2}$$

where $Y_i$ is the SOA yield of $VOC_i$, which can be initially estimated according to the OA concentrations and VOC/NOx conditions using two-product models summed up by previous chamber studies (Lim and Ziemann, 2009; Ng et al., 2007). For species lacking yield curves, the fractional aerosol coefficient (FAC) values proposed by Grosjean and Seinfeld (1989) were used. Furthermore, the yields were corrected for vapor wall losses according to a recent study (Zhang et al., 2014). The SOA yields estimated in this study and the corresponding references are shown in the supplement as Table S3. $VOC_i$ and $VOC_{i,\ consumed}$ represent the initial emission and the real amount of $VOC_i$ oxidized by the OH radical, which can be calculated with the photochemical age (de Gouw et al., 2005; Warneke et al., 2007):

$$VOC_i = VOC_{i,t} \times \exp(k_i[OH]\Delta t) \tag{3}$$

$$VOC_{i,consumed} = VOC_i - VOC_{i,t} \tag{4}$$

where $[OH]\Delta t$ represents the OH exposure, which can be estimated based on the ratio of (MACR+MVK)/isoprene according to previous studies (Santos et al., 2018; Apel et al., 2002; Stroud et al., 2001; de Gouw et al., 2005). Accordingly, the OH-driven isoprene oxidation and the sequential reaction model can be described as:

$$Isoprene + OH \rightarrow 0.54 \times (MACR + MVK) \tag{5}$$

$$k_I=1\times10^{-10}\ cm^3\ molecule^{-1}\ s^{-1}$$

$$(MACR + MVK) + OH \rightarrow Product \tag{6}$$

The derived expression can be used to estimate the photochemical age, which is shown as follows:

$$\frac{[MACR + MVK]}{[Isoprene]} = \frac{0.54k_1}{(k_2 - k_1)}\left(1 - e^{(k_1-k_2)[OH]\Delta t}\right) \tag{7}$$



**2.5 Radical contribution quantification**

Hydroxyl radicals (OH) are the most important oxidants in the troposphere atmosphere, which mainly originate from the photolysis of $O_3$, OVOCs, and HONO (Crutzen et al., 1991). Except for in the morning, carbonyl compounds and $O_3$ are major precursors of OH. In this study, the production rates of OH from $O_3$ and typical carbonyls (formaldehyde, acetaldehyde, and acetone) were calculated using the following formulas:

$$P\left(OH_{O_3}\right) = \frac{2J_{O_3}[O_3]k_1[H_2O]}{k_1[H_2O] + k_2[O_2] + k_3[N_2]} \qquad (8)$$

$$P\left(OH_{Carbonyl_i}\right) = 2 \times J_{i,l}[Carbonyl_i] \qquad (9)$$

where $J_{O_3}$ and $J_{i,l}$ represent the photolysis rate constants of $O_3$ and carbonyl compounds, respectively, which are obtained from the TUV model. The model calculates the solar shortwave radiation between 120 and 735 nm with most parameters set at default values, except for the aerosol optical depth (AOD). The AOD is set as 0.7 for non-haze days and 1.5 for haze days, as in previous studies in Beijing (Liang et al., 2013; Rao et al., 2016). In formula (8), $k_1$, $k_2$, and $k_3$ represent the reaction rate constants of O (1D) with $H_2O$, $O_2$, and $N_2$, respectively, which are obtained from the National Institute of Standard and Technology (NIST) Chemical Kinetics Database (https://kinetics.nist.gov/kinetics/, Standard Reference Database 17, Version 7.0). It needs to be noted that in Eq. (9), all the $RO_X$ produced from the photolysis reaction can be assumed to be completely converted into HO, which is the case in most periods during the daytime. Therefore, the results show the upper limit of the carbonyl contribution (Wu et al., 2017; Zheng et al., 2013). Furthermore, the yield of radicals (Y) of a carbonyl from the main reactions in the daytime can be calculated as:

$$Y = \frac{2\sum_{i=1}^{n} J_{i,1}[OVOC]_i}{\sum_{i=1}^{n}(J_{i,1} + J_{i,2} + \cdots\cdots + J_{i,m})[OVOC]_i + \sum_{i=1}^{n} k_{i,OH}[OH][OVOC]_i} \qquad (10)$$

where $J_{i,2-m}$ represents the photolysis rates without the formation of OH radicals, which is also estimated with the TUV model. In addition, $k_{i,OH}$ is the rate constant for the reaction of carbonyl with OH radicals. [OH] is set as $0.62 \times 10^6$ molecules $cm^{-3}$ on





clear days and $0.46 \times 10^6$ molecules cm$^{-3}$ on hazy days, calculated using the following
equation, based on a parameterization method (Ehhalt and Rohrer, 2000).
$$[OH] = 4.1 \times 10^9 \times (J_{O1D})^{0.83} \times (J_{NO2})^{0.19} \times \frac{140 \times [NO_2] + 1}{0.41 \times [NO_2]^2 + 1.7 \times [NO_2] + 1} \quad (11)$$
Here, $J_{O1D}$ and $J_{NO2}$ are the estimated photolysis frequencies (s$^{-1}$) of O$_3$ and NO$_2$
with the TUV model, respectively. [NO$_2$] is the measured NO$_2$ concentration (ppbv,
parts per billion by volume).
**3.  Results and discussion**
**3.1  Mixing ratio of VOCs**
As previously described, PTR-MS is highly efficient for detecting OVOCs and
benzenoids but useless for detecting alkanes because of the limitation of the proton
affinities. Therefore, the composition of VOC species is initially summed up by
integrating the observation data from PTR-MS and canister sampling. The concordance
of the observed data from the two measurements is checked using the species that can
be detected from both methods, and the results show a good relationship between the
two methods for benzene (r=0.90). On average, the observed benzene concentration
from PTR-MS is 12% higher than those from canister sampling. Figure 1 shows the
average mixing ratios of different VOC categories during the sampling period,
including alkanes (>C6), benzenoids, alkenes, halogenated hydrocarbons, nitrogen-
containing compounds (N-VOCs), sulfur-containing compounds (S-VOCs), and
OVOCs. Accordingly, the mixing ratio of OVOCs is more than 5-fold that of the others.
Among the major radical contributors—carbonyl compounds—acetaldehyde has the
highest average concentration, followed by a mixture of acetone and propionaldehyde,
as well as formaldehyde. In addition, alcohols such as methanol and ethanol exhibit
high levels, which might be largely due to the combustion process (Sahu and Saxena,
2015; Holzinger et al., 2005; Brito et al., 2015). Benzenoids are the second most
abundant category, and the major components are benzene, toluene, and C8 aromatics.
Under the current instrument configuration, short-chain alkanes and unsaturated
hydrocarbons cannot be detected effectively. The alkanes with more than two carbon
atoms have an average mixing ratio of 1.47 ppbv and are mostly cycloalkanes, heptane,



and octane. The alkenes contain isoprene, pinenes, and styrene, with a total
concentration of 1.61 ppbv. In addition, the observed N-VOCs and halohydrocarbons
have mixing ratios of 2.55 and 2.64 ppbv, respectively. Finally, only $CS_2$ and dimethyl
sulfide can be detected for the S-VOC category, with a total concentration of 0.62 ppbv.

Figure 2 shows a comparison between the mixing ratios of carbonyls and

benzenoids measured in Xinxiang, with the observed results reported elsewhere.
Overall, the average VOC concentrations are among the highest levels worldwide.
Formaldehyde has been regarded as the largest radical contributor in winter, while the
average concentrations in Xinxiang on both hazy days and non-hazy days are lower
than those of most recent studies in China (Qian et al., 2019; Sheng et al., 2018; Yang
et al., 2019; Su et al., 2019). In contrast, the average concentrations of acetaldehyde
and C3 carbonyls on both hazy and non-hazy days are commonly higher than those in
previous studies. The concentrations of benzenoids are generally above the worldwide
level except for some severe pollution periods in China and India (Sahu et al., 2016;
Sheng et al., 2018; Liu et al., 2015; Sinha et al., 2014). In autumn and wintertime, the
above carbonyl compounds and benzenoids are mainly from anthropogenic emissions
(Qian et al., 2019; Yang et al., 2019; Barletta et al., 2005; Zhang et al., 2015). Therefore,
poor diffusion conditions and intensive emissions are the major reasons for the high
concentrations of VOCs on hazy days. It is obvious that the formaldehyde, toluene, and
C8 aromatics show slighter enhancements from non-hazy days to hazy days, probably
because of the fast consumption owing to their high photolysis and photochemical rates.
**3.2 Meteorological conditions and diurnal variations**

It has been shown that pollutant dispersion is primarily related to wind direction

and speed, while atmospheric chemical reactions can be influenced by temperature and
humidity (Greene et al., 1999). In Fig. 3, the hourly averaged data of the meteorological
parameters and the concentration of pollutants including OVOCs, benzenoids, and
some inorganic gases are displayed. During the investigated period, the temperature
and RH show opposite diurnal fluctuations, and both show significant changes after
entering the second half of November. In particular, RH remained high on 12-14



November and 1-3 December, which might provide a preferable environment for
sulfate and OA increases (Jathar et al., 2016; Sun et al., 2013; Tang et al., 2018). It is
evident that the variation trends of the pollutant concentrations are highly ascribed to
atmospheric diffusion characterized by wind speed and PBL height. The first peak of
benzenoids, on 9-10 November, appears after a valley of PBL and dozens of hours of
low wind speed. During 12-15 November, the high concentration levels of both OVOCs
and benzenoids occur with sustained low wind speed (< 2 m/s), and this state is
dispersed with the increased wind speed after 12:00 on 15 November, along with a
significant increase in PBL height. In the last period of haze (especially after 23
November), the VOC concentrations fluctuate at a high level, and the wind direction is
more concentrated in the northeast at speeds of less than 3 m/s. Due to the
implementation of motor vehicle restrictions, the benzenoid concentration during this
period is lower than that on the last hazy days. An investigation in northern China
proposed that high pressure followed by a low-pressure system and front zone is in
accordance with the increasing phase, maximum values, and decreasing phase of
pollution (Chen et al., 2008), and this pressure pattern can be clearly observed and used
to explain the accumulation process on 7-9, 16-20, and 24-26 November. The
meteorological conditions on 1-3 December accord with both poor diffusion conditions
(shallow PBL) and accumulation pressure patterns; therefore, pollutants such as
OVOCs, benzenoids and CO show an increasing tendency afterwards. It should be
noted that the variation trend of $SO_2$ does not agree with the others, especially on 1-3
December. This phenomenon shows that the major source of $SO_2$ is quite different,
which might be due to long-distance transportation.
Prior to the investigation into the variation characteristics of typical carbonyls and
benzenoids, daily changes in temperature, RH, wind speed, and air pressure are
demonstrated in Fig. S3 to distinguish the influences of meteorological factors. In
addition, the diurnal changes in some inorganic gases (NOx, $SO_2$, and CO) are
investigated as tracers of typical sources, as shown in Fig. S4. The detailed analysis
referring to the diurnal variation law of meteorology and tracers is described in Text S5.





Briefly, atmospheric pollutants tend to accumulate in the morning and disperse in the
afternoon along with changes in atmospheric stability. $NO_X$ and CO show the bimodal
features of vehicle emissions, and $SO_2$ represents the regional influence of coal
combustion. According to Fig. 4, the concentrations of all the carbonyl species begin
to climb after sunrise and reach high concentrations around noon, which might partly
result from secondary photochemical production. Meanwhile, industrial activities are
completely resumed during the daytime, contributing to the above concentration pattern.
The right Y-axis in Fig. 4 represents the ratio of the concentration to the value at 0:00
LT. Comparatively, the increase ratio of the C3 carbonyl mixture is the highest,
followed by those of acetaldehyde and formaldehyde, which has an inverse correlation
to the photolysis rates among formaldehyde, acetaldehyde, and acetone. Therefore,
photochemical consumption plays an essential role in carbonyl compound behaviour in
the atmosphere, and the mixture of C3 carbonyls may be dominated by acetone. The
correlation coefficient between benzene and CO is 0.92, and their diurnal variation is
quite similar. The reaction activity of toluene in photochemistry is strong; therefore, the
concentration after 12:00 LT decreases sharply. Figure 4 (f) indicates that C8 aromatics
are mainly emitted from solvent usage, and the temperature influence on the
evaporation rate can offset the daytime "valley" caused by meteorological conditions
in the afternoon.
**3.3  Source apportionment of VOCs**
Figure 5 shows the correlation between the estimated total VOC concentrations
and the measured total VOC concentrations with the coefficient r=0.96, indicating that
the source apportionment result fits well with the measured data. The profiles of the
sources resolved by the PMF model are shown in Fig. S7, including biogenic and
secondary sources, solvent evaporation, residential heating, thermal power generation,
vehicle exhaust, and industrial emissions. Details of the source identification are
described in Text S6 on the basis of the corresponding tracers and daily variation
(shown in Fig. S8). Generally, benzenoids are closely related to coating, solvent usage,
industrial processes, and energy structure; therefore, it is difficult to compare the



contributions among different seasons and cities in the world (Hui et al., 2018; An et
al., 2012). In this study, solvent evaporation is the largest contributor, with a fraction
of 47% (shown in Fig. 6 (a)), followed by residential heating (19%) and industrial
emission (16%), and this situation occurs because the sampling site is located in a
district with industrial zones and restricted traffic during the sampling time.
Anthropogenic emissions are found to be the largest contributor of OVOCs
(specifically carbonyl compounds) in the current study, which aligns with the previous
conclusions about winter (Qian et al., 2019; Chen et al., 2014). Figure 6 (b) shows the
distribution of all OVOC emissions from the six sources, and the proportions are close.
For individual species, 64% of formaldehyde, the most reactive species for radical
production, is contributed by primary anthropogenic emissions, including thermal
power plants (19%), industrial emissions (16%), solvent evaporation (11%), and other
sources. The main sources of acetaldehyde and C3 carbonyls have been reported as
vehicle exhaust, industrial emissions and solvent usage (Qian et al., 2019; Singh et al.,
1994), and industrial emission is found to be the largest contributor in this study.
**3.4 Regional contribution**
In addition to the impact of local emissions, regional transport is considered in this
study. As shown in Fig. 7, four clusters exist during the sampling time. In total, the
dominant air masses are from the southwest direction, including long-distance transport
mainly from western Inner Mongolia (20.0%) and medium-distance transport from
Shaanxi and Shanxi Provinces (34.2%). The proportion clusters from the northeast
(passing through the BTH region) and southeast (from southern Henan province and
some areas in Hubei, Anhui, and Jiangsu province) are 21.5% and 24.3%, respectively,
with a relatively shorter distance. According to the PSCF and CWT results in Fig. 8 and
Fig. 9, the VOC concentrations are dominated by local emissions and can be influenced
by cities within Henan Province or in neighbouring provinces. As biogenic emissions
are extremely low in northern China in autumn and BVOCs generally actively react
within a few hours, Fig. 8 (a) and Fig. 9 (a) represent the spatial origins of the secondary
products. The intensity of this source has a relatively high value in northern Shanxi, as





393 well as at the junction of Hebei and Shandong Provinces, which might be influenced

394 by the oxidation of VOCs from other sources. According to Fig. 8 (b) and Fig. 9 (b),

395 hot spots of solvent evaporation are found to the west of Xinxiang. Intensive residential

396 heating emissions (shown in Fig. 8 (c) and Fig. 9 (c)) are also found to the west of

397 Xinxiang, covering the border cities in Henan and Shanxi, which might be due to the

398 high coal production and residents' living habits. Another coal combustion-related

399 source, thermal power generation, has relatively high PSCF and CWT values around

400 the sampling site, with some hot spots observed in the southern region. In Fig. 8 (e) and

401 Fig. 9 (e), the contribution potential of vehicle exhaust is slightly higher from the

402 northwest and source directions. The industrial emission sources are obviously

403 observed around the sampling site with some hot spots except for to the west, which

404 coincides with the real situation, where many plants are located in the remote districts

405 around the urban area.

**3.5 Potential roles of VOCs in SOA formation**

407   The ratio of VOCs/NOx (ppbC/ppbv) is calculated with the species contributing

408 to SOA formation, and the value is generally below 5 during the investigated period,

409 indicating high $NO_X$ conditions. Based on the estimated yields of the VOCs shown in

410 Table S3, the SOAFPs are calculated and compared in Fig. 10. It should be noted that

411 the yields of mixtures of isomers are estimated based on the yields of individual species

412 and their weighted concentrations (estimated from canister sampling results).

413 Accordingly, aromatics have the largest SOAFP, with contributions of 66%, 7.2%, and

414 4.5% from toluene, benzene, and C8 aromatics, respectively. The estimated SOAFP

415 percentage of alkanes is low in the current study (~2%). Although short chain alkanes,

416 especially species with fewer than 2 carbon atoms, are not detected in this study; this is

417 not the reason for the low contribution proportion because many previous studies have

418 shown that the yields of alkanes (<C7) are close to zero (Grosjean and Seinfeld, 1989;

419 Gao et al., 2019). Compared to a recent study in a nearby city, Zhengzhou, the observed

420 species of alkanes (>C2) are similar and are mainly emitted from petrochemical sources

421 and oil gas evaporation. To our knowledge, these kinds of sources are not intensive in

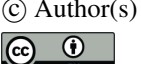



Xinxiang. Long-chain alkanes (>C12) are intermediate volatile organic compounds (IVOCs) that are of great significance in SOA formation, but they cannot be observed using the current method, which would be the main reason for the underestimation of alkane contributions. The SOAFP contribution of OVOCs in this study (2%) is close to the inventory-based SOAP estimation in the Beijing-Tianjin-Hebei region (1.9%) (Wu and Xie, 2018), which is dominated by some long-chain alcohols and phenols.

Based on ACSM analysis, OOA (approximately treated as SOA in this study) accounts for 63±10% of OA during the sampling campaign (shown in Fig. S1). As previous studies described, SOA can be formed via gas-phase oxidation of POA and VOCs, as well as some heterogeneous reactions on aerosol surfaces (Xing et al., 2019). To investigate the proportion of SOA generated from VOC photooxidation in gas phases along with the pollution process, real-time consumption and SOA production of benzene, toluene, and C8 aromatics are calculated based on Eqs. (2)-(7). As shown in Fig. 11, the weights of the benzenoids in SOA formation decline dramatically as the OOA concentration increases from 2 to 5 μg m$^{-3}$, and the proportions of toluene, C8 aromatics, and benzene are approximately 1%, 0.5%, and 0.1%, respectively, with high OOA concentrations. Considering that benzenoids are the precursors with the largest SOAFP, it can be concluded that the direct oxidation of VOCs significantly contributes to SOA accumulation at the initial pollution, but the weight declines with the aggravation of haze.

Figure S9 shows the correlation curve of OH exposure versus OA concentration with a correlation coefficient (r=0.65), indicating the identity of the atmospheric oxidation capacity with SOA increase. In Table 1, the modelled average OH concentrations, OH production rates of O$_3$ and typical carbonyls, and the radical yields of carbonyl compounds on both non-hazy days and hazy days are listed. The estimated OH concentration is close to that of a previous study in Beijing (0.99×10$^6$ molec cm$^{-3}$ on non-hazy days and 0.34×10$^6$ molec cm$^{-3}$ on hazy days (Rao et al., 2016)), while the difference in this study is smaller, which might be because of the larger accumulation of photolysis precursors on hazy days in Xinxiang. Among the VOCs, formaldehyde is



the largest radical contributor, but the radical production rate decreases by 9% on hazy
days, resulting from the reduction in luminous intensity. As described in Sect. 3.1,
acetaldehyde concentration is at a high level worldwide; therefore, its contribution to
radical production is significant and shows a 21% increase from non-hazy days to hazy
days. As acetone and propionaldehyde cannot be distinguished in the current method,
two radical production rates are listed assuming that only a single species exists.
Accordingly, the above two species have significantly different photolysis properties,
which results in some uncertainties in the calculations of the total radical production
rate and yield. Therefore, the radical yields are given as ranges indicating the probable
number of OH radicals generated when one carbonyl compound molecule is consumed
in the ambient air. During the investigated period, the yields are estimated to be within
0.38-0.43 under different conditions. The contribution of radicals from $O_3$ photolysis is
small in wintertime, and little difference is observed between non-hazy days and hazy
days.
Finally, the contributions of SOAFP and radical production among the resolved
source are demonstrated, which are calculated with the concentration of every VOC
species in each source and the relevant yields or photolysis rates. According to Fig. 6
(c), the top 2 contributors are solvent evaporation and industrial emission, followed by
residential heating and vehicle exhaust. Similarly, the averaged $RO_X$ production rates
of each source are calculated with their resolved carbonyl concentrations and the
corresponding photolysis rates weighted by 16 non-hazy days' and 11 hazy days' values.
It should be explained that the photolysis rates of C3 carbonyls used in this section are
the average values of acetone and propionaldehyde. As shown in Fig. 6 (d), the potential
major sources of radical production are solvent evaporation, biogenic and secondary
production, industrial emissions, and vehicle exhaust.
**4. Conclusion**
VOCs are essential in SOA formation because of their dual roles as precursors and
oxidant producers. In this study, online measurements of NMVOCS are conducted for
the first time in urban Xinxiang, with some canister samples grabbed for supplemental



information. Overall, OVOCs are the category with the highest mixing ratio, which is
more than 5-fold higher than those of other categories. Herein, methanol, acetaldehyde
and C3 carbonyls are among the most abundant species. Benzenoids are the second
most abundant category, with benzene, toluene, and C8 aromatics as major components.
Secondary production is significant in this study, as the growth amount of OVOCs is
relatively larger than that of benzenoids on hazy days, which coincides with the source
analysis results. Compared with the other field studies in the world, the mixing ratio of
benzenoids such as benzene, toluene, C8 aromatics, as well as OVOCs including
acetaldehyde, and C3 carbonyls are at high levels, indicating intensive anthropogenic
emissions in Xinxiang. In total, six sources were resolved in this study, including
biogenic and secondary sources, solvent evaporation, residential heating, thermal
power generation, vehicle exhaust, and industrial emissions. Accordingly, benzenoids
are highly loaded with solvent evaporation, residential heating, and industrial emissions,
while the six resolved sources have similar contributions of total OVOCs. According
to the PSCF and CWT results, ambient VOCs are more likely influenced by local
emissions and short-distance transport. In brief, secondary products are transported
along with the air mass movement trajectory, while solvent evaporation and residential
heating sources are intense around the border cities in Henan and Shanxi Provinces. In
addition, thermal power generation, vehicle exhaust, and industrial emissions are
intensively released around the sampling site, with some hot spots in different directions.

Based on the SOAFP calculations, toluene and benzene are the top 2 SOA

contributors in the campaign, while the weights of their estimated production mass in
the observed SOA mass show a declining trend with the aggravation of pollution. On
the other hand, the OH exposure and SOA concentration exhibit a good relationship
throughout the observation period, which highlights the effect of atmospheric oxidation
capacity on SOA growth. From the estimation of the OH radical production rates,
formaldehyde is found to be the strongest radical contributor, while the contribution of
acetaldehyde is significant because of its high concentration in this study. Isomers such
as acetone and propionaldehyde cannot be distinguished in PTR-MS; therefore, the



radical production rate of C3 carbonyls is presented as a range. Because the photolysis
properties of the above isomers differ greatly, further analysis should be performed to
clarify the proportion of C3 carbonyls and their OH contributions. In this study, we
assigned SOAFP and radical production information into the resolved sources. The
results show that solvent evaporation is the dominant source for SOAFP and radical
production. Comparatively, residential heating has a greater impact on SOAFP, and
industrial emissions are more significant for radical production. Although traffic
restriction is conducted during the campaign, the influence of vehicle exhaust is still
non-neglectable in the two aspects. In addition, biogenic and secondary sources have
large potential in radical production, which could not be undervalued in autumn or even
in winter. This study provides first-hand information on VOC characteristics in a central
Chinese city and first evaluates the resolved sources with SOAFP and radical
production information. The results will provide targeted measures for SOA reduction.

*Data availability*. The observational data in this study are available from the authors
upon reasonable request (zhanghaixu@tsinghua.edu.cn).

*Author contributions.* HZ designed the experiments, carried out the field observation,
processed the data, and drafted the paper with contributions from all co-authors. CC
maintained the instruments during the campaign, processed ACSM data, and conducted
source apportionment analysis of organic aerosols. WY and NW assisted setting up
field campaigns and sampling. YB coordinated the field campaign. QZ and KH
supervised the project and revised the paper.

*Competing interests*. The authors declare that they have no conflict of interest.

*Acknowledgements.* This work was funded by the National Research Program for key
issues in air pollution control (DQGG0201) and National Natural Science Foundation
of China (41625020 and 41571130035). The authors would thank Xinxiang Ecological



Environmental Bureau for logistical support on the filed campaign.

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








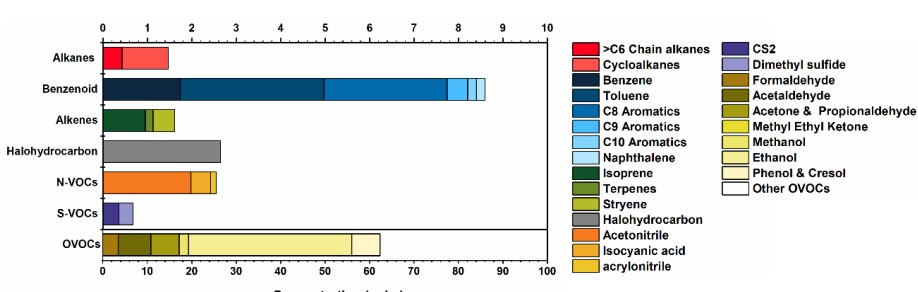


Figure 1. The average concentrations of VOCs observed in Xinxiang.





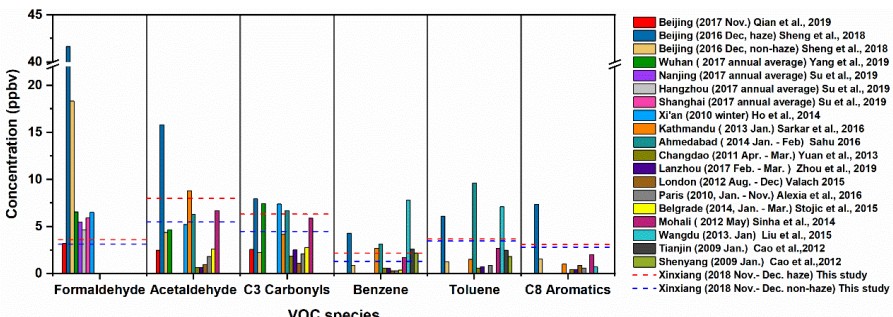


Figure 2. Comparison of average concentrations of OVOC and benzenoids in Xinxiang
Valley (hazy and non-hazy periods) with VOC levels at other sites worldwide.



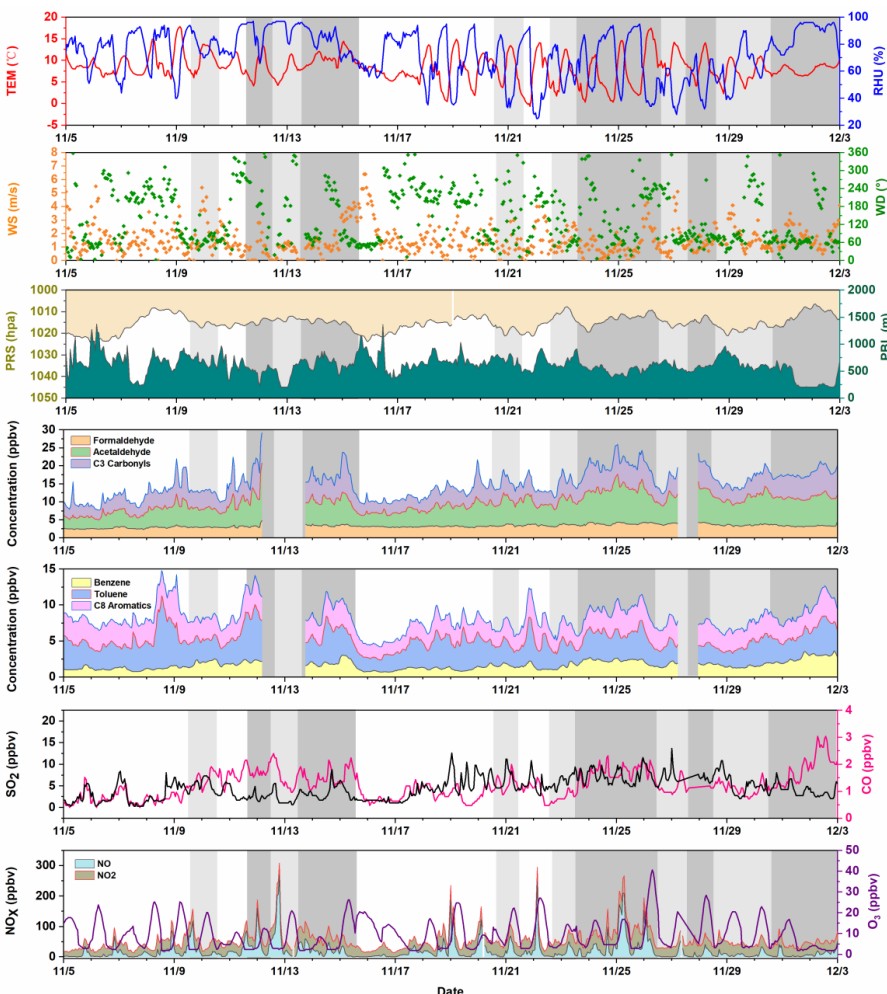


Figure 3. Time series of meteorological parameters and pollutant concentrations in
Xinxiang during the sampling period. The colours of the background represent the daily
averaged concentration levels of $PM_{2.5}$, and white, light grey, and dark grey represent
0-75 $\mu g\ m^{-3}$, 75-115 $\mu g\ m^{-3}$, and >115 $\mu g\ m^{-3}$, respectively.






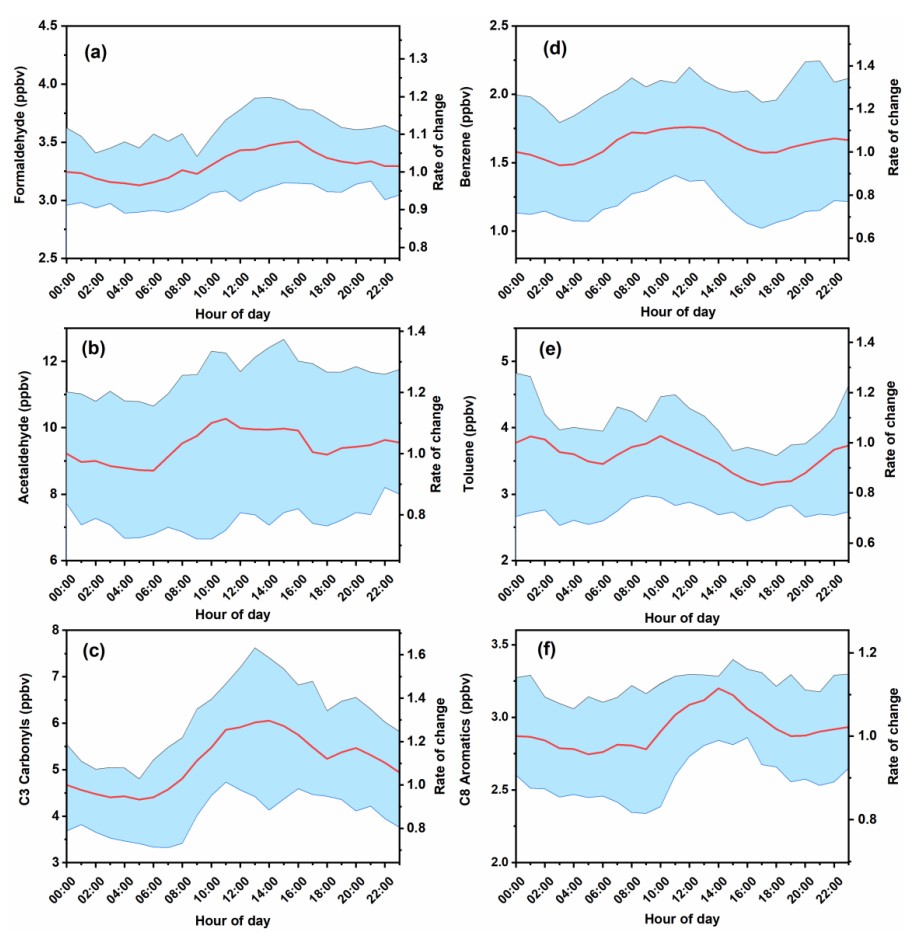


Figure 4. Diurnal variations of formaldehyde (a), acetaldehyde (b), C3 carbonyls (c),
benzene (d), toluene (e), and C8 aromatics (f).





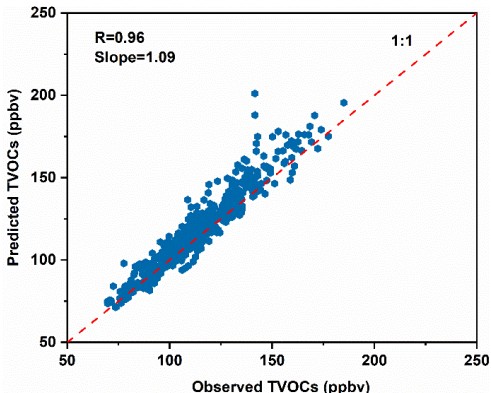


Figure 5. Correlation between observed and estimated VOC concentrations from the
PMF model.





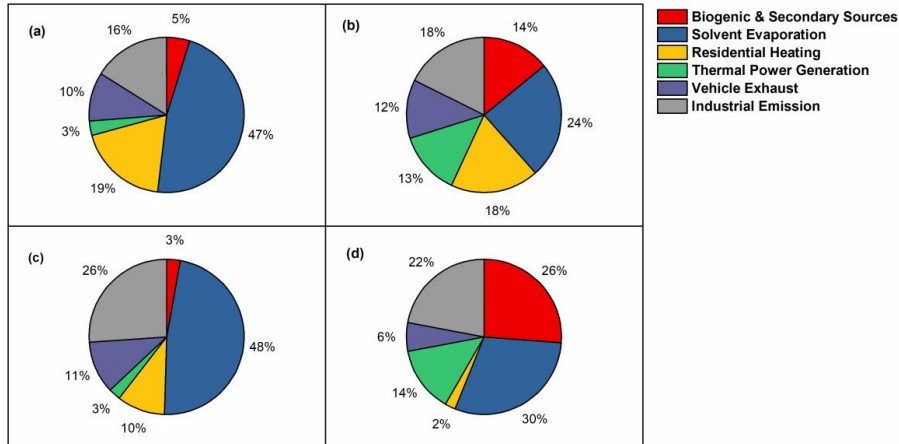

Figure 6. The contributions of different sources derived from the PMF model to benzenoids (a), carbonyl compounds (b), SOAFP (c), and radical generation rate (d).



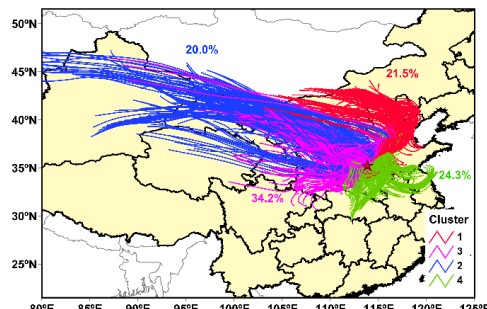

Figure 7. Cluster analysis of 48 h backward trajectories for Xinxiang from 2018.11.05
to 2018.12.03, with the percentage of each cluster presented.






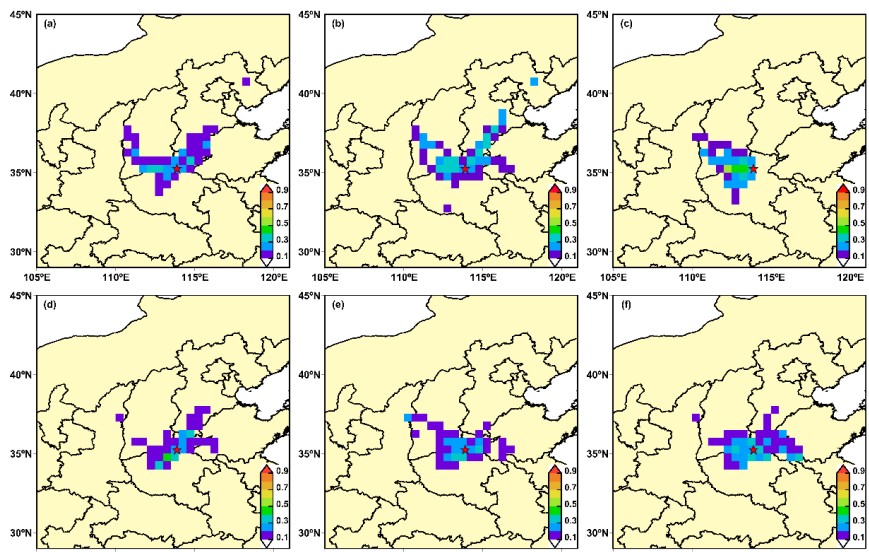


Figure 8. Weight potential source contribution function (WPSCF) maps for identified
sources derived from PMF analysis including biogenic and secondary sources (a),
solvent evaporation (b), residential heating (c), thermal power generation (d), vehicle
exhaust (e), and industrial emissions (f).



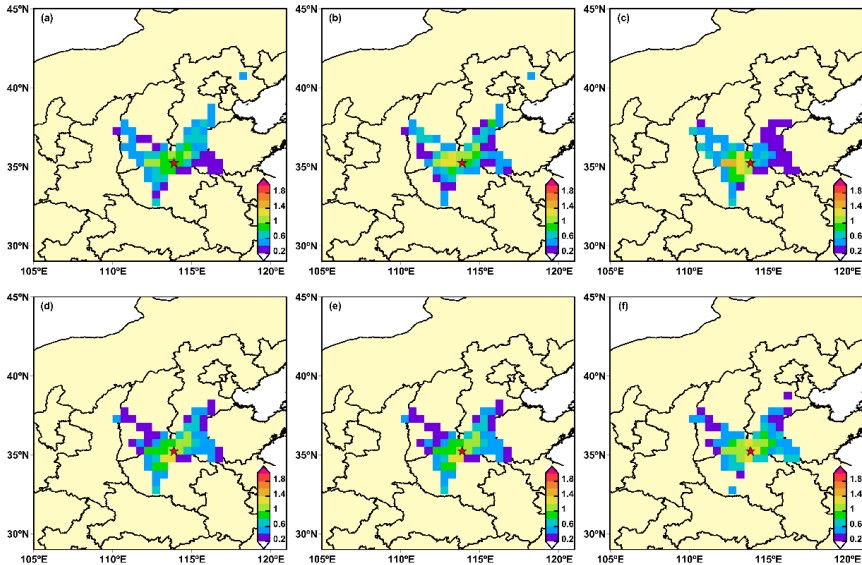


Figure 9. Weight concentration-weighted trajectory (WCWT) maps for identified sources derived from PMF analysis including biogenic and secondary sources (a), solvent evaporation (b), residential heating (c), thermal power generation (d), vehicle exhaust (e), and industrial emissions (f).






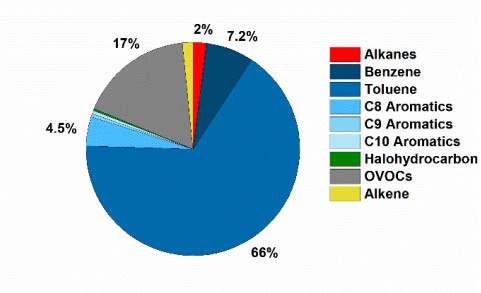


Figure 10. Chemical distribution of SOAFP during the sampling period in Xinxiang.






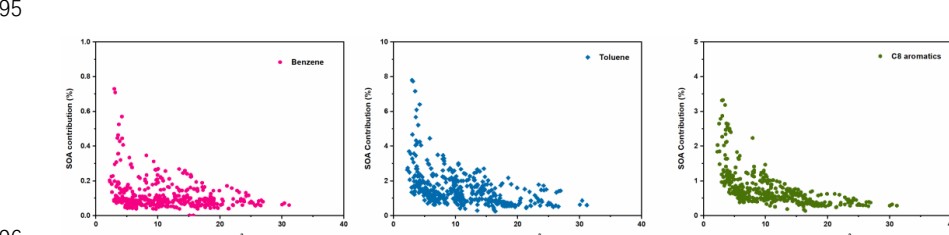


Fig. 11 SOA contributions generated by benzene (a), toluene (b), and C8 aromatics (c)
with increasing OOA concentration.





**Table 1. The modelled daytime (8:00-16:00 LT) average OH concentration ($\times 10^6$ molec cm$^{-3}$)**
and **radical production rate (P, $\times 10^6$ molec cm$^{-3}$ s$^{-1}$) of O$_3$ and carbonyls and OH radical yield**
**(Y) from photolysis of carbonyl compounds.**

|  | OH concentration | $P_{O3}$ | $P_{HCHO}$ | $P_{CH3CHO}$ | $P_{C2H5CHO}$ | $P_{(CH3)2CHO}$ | Y |
|---|---|---|---|---|---|---|---|
| Non-hazy days | 0.62 | 0.07 | 1.31 | 0.24 | 0.85 | 0.01 | 0.39-0.43 |
| Hazy days | 0.46 | 0.07 | 1.19 | 0.29 | 0.95 | 0.01 | 0.38-0.40 |
