# Peer review of "Measurement report: Characteristics and sources of non-methane"

_Atmospheric Chemistry and Physics, 2020_

## Referee Comment (RC1) · Anonymous Referee #1 · 24 Apr 2020

The authors describe result of a field campaign in a city in Central China, in which VOCs were measured by online PTR-MS and by GC from canister samples. Additional information about meteorological parameters and inorganic compounds is given. The aim of the paper was to characterize sources of VOCs and their role for SOA formation. The design of the study only allowed observing a rather incomplete picture concerning the goal of the study. The limitations of the study and consequences for the results, however, are insufficiently discussed in the paper. The authors try to calculate various quantities such as chemical age of air masses, OH concentrations at the location of measurements, radical production rates using approaches that cannot be easily justified to be applicable for conditions of the campaign and with the limited number of

measurements that were done. Overall the paper reads as if the authors mainly try to calculate various parameters from measured values in a similar way as reported in literature, but without discussing the science behind. The main part of the paper is describing VOCs measurements, but does not give any new results about chemical processes or state of the atmosphere. VOC data are further analysis with respect to source of VOCs by a PMF analysis, which can be done with the limited number of measurements. Overall, however, the content of the paper is mainly a description of the level of a limited number of VOC concentration that were measured for one month in a city in China and attributing these VOCs to sources that are to be expected. Therefore, the paper does not contain sufficient new results to be published in ACP. In addition, scientific quality of the manuscript is not good enough, because limitations of applied methods are not discussed.

Some specific comments:

Radical sources: The authors claim that photolysis of OVOCs is the major source of radicals. This has been shown for a specific environment in the center of oil- gas-fields in the US, but cannot be generalized. The limited number of measurements in the campaign described in this paper does not allow to analyse source of radicals, neither appropriate radiation measurements were done nor important radical precursors such as nitrous acid were detected. Though mentioned in the text, the authors continue stating that carbonyl photolysis yields OH (Eq 9), which is not true.

OH concentration: The authors use a parameterization for calculating OH concentrations, for which important parameters like photolysis rates are only estimated. However, even if they had measurements, this approach is not applicable, because the parameterization was achieved for a totally different environment and location. There cannot be expectation that this parameterization can be used to calculate OH for the conditions of this campaign. Chemical age: The authors use the ratio of isoprene to MVK+MACR to estimate the OH dose that give the chemical age of the air mass. This approach would only be applicable, if the air mass contained VOCs from approximately

co-located emission sources that are transported to the measurement site without significant mixing of air masses from other sources. As the authors show there is little biogenic sources, but the majority of VOCs stem from anthropogenic sources. Therefore, it cannot be expected that the chemical age of anthropogenic sources can be estimated by using the degradation of isoprene which is emitted by plants. The location of the measurements is in a city, where anthropogenic sources of VOCs are presumably very close to the site. As a consequence, also the estimate for SOA production is based on questionable assumptions.

VOC measurements: The authors do not discuss consequences of the limited number of VOC species that were detected. For example, small alkenes, which often make a larger contribution to the total number of reactants in anthropogenic environments like here, are missing. Concerning the data quality there is only good agreement between PTR and GC measurements for benzene stated, but nothing said, if good agreement was also found for other species. Figure 2 gives a comparison of VOCs concentrations with other locations, but no conclusions can be drawn, because VOC levels highly depend on the distance to sources, time of the year etc. The authors do not make an attempt to give any interpretation, when they compare their VOC levels with those found in other locations. Figure 5 gives a correlation between observed TVOC and predicted TVOCs. It is insufficiently explained how predictions of TVOCs were derived, but it seems as if predictions rely on the PMF analysis. If this was the case, a good correlation is certainly expected, because the PMF factors themselves are based on measured VOCs. The effect of prediction would only be smoothing out some of the variability of concentrations of single VOCs.

Technically: The quality of figures is poor due to small font sizes, small sizes of bars in bar plots and small legend sizes.

---

## Referee Comment (RC2) · Anonymous Referee #2 · 31 May 2020

Zhang et al. performed online and offline VOC measurements using a PTR-MS and canister samples, respectively, in a central city in China. They compare their measurements to historical data, perform a factor analysis to identify primary VOC sources, and comment on the potential of the VOCs to form SOA. Based on a preliminary analysis, they find BTEX and C3 carbonyls concentrations to be elevated compared to measurements elsewhere, solvent evaporation to be an important source for the measured VOCs, and benzenoid species to be important for SOA formation in the urban environment.

There is a clear need for detailed online VOC measurements, on which analysis similar

to those performed in this work would lead to an understanding of the primary sources to air pollution in populated cities. Hence the work is well motivated and the measurements are likely to be valuable to the atmospheric community. However, the analyses performed on the data are insufficiently described in terms of the methods and results. I generally recommend publication of this manuscript after the authors have had a chance to review my broad comments and responded to it.

Overall comments:

1. Measurement report: This article is submitted as a 'measurement report' but it does not seem to follow the data policy outlined on the ACP website: 'The data presented in measurement reports must be openly accessible in accordance with the EGU data policy.'. The data availability section in this paper currently reads 'The observational data in this study are available from the authors upon reasonable request (zhang-haixu@tsinghua.edu.cn).'. Please make sure that the data are available before final submission.

2. Data analysis: My understanding is that the measurement reports 'may include model results and conclusions of more limited scope than in research articles'. The authors, however, have begun to undertake a modest amount of analysis on the VOC measurements to study the source contributions, regional dependence, and implications for SOA. However, I do not think the analysis methods are presented clearly enough and/or the analysis results are described in detail for me to consider them complete; more detailed comments below. My interpretation of 'limited scope' in the context of this paper is to perform a 'complete' analysis for one of the tasks (e.g., source apportionment based on a PMF analysis) rather than a 'light' analysis for many tasks. In case this article is published with its current structure, it should be made clear that the analyses presented in this work are preliminary and will be continued to be worked on before they are published in their final form as a companion research article.

3. PTR-QMS: I know of a few earlier studies that have used a quadrupole PTR-MS

to measure VOCs in urban environments (e.g., Warneke et al. (2012)). As the VOC species are measured at a unit mass resolution in a PTR-QMS, are the species identified and quantified in this work free from interferences? Depending on the answer to the interference question, can the data be used to perform PMF or would the interferences confound the source apportionment? For the reader's sake, can the authors cite earlier work that have used the PTR-MS measurements in the same manner as they are used here and state any limitations in performing such analysis? In addition, I am not sure how secondary production, in general, is resolved in PMF when both primary and secondary species are present in the same factor and when species within a factor have different reactivities.

4. Confusing phrases: Throughout the manuscript, there are lots of instances where it was very hard to understand what was being said. The technical communication needs to be substantially improved in the revision.

Detailed technical comments:

1. Line 23: what does 'obviously' mean?

2. Line 26: what does 'high levels' mean? A recurring comment is that the authors describe trends qualitatively, which doesn't help with the interpretation.

3. Line 28: Are benzenoids only reduced single-ring aromatics in this work? If yes, please specify that or use 'single-ring aromatics'.

4. Line 58: '20-30%' of what? Another recurring comment is the text does not provide context for what is increasing or decreasing and the reader is left guessing (sometimes incorrectly).

5. Lines 68-71: That the 2-product or n-product (i.e., volatility basis set) SOA model has not been used to interpret field data is misleading. There is extensive literature over the past two decades that points to the use of 2/n-product models for field analysis (e.g., Dzepina et al., 2009).

6. Line 81: What are these 'differences in different regions'? Are there some universal trends in ranking of the sources?

7. Line 93: What is 'autumn'? Why not just specify the dates?

8. Line 123: 'as focus or tracers are'? Please correct.

9. Sections 2.2-2.3: My general sense is that the detailed provided in these sections is not sufficient, especially if the reader is not familiar with the techniques.

10. Line 187-188: My sense is that PSCF and CWT need to be redefined in the text, separate from the abstract.

11. Lines 210-220: The other technique I have seen is the toluene/benzene ratio. Was there a reason why isoprene, MACR, and MVK were used to deduce OH exposures? Also, are MACR and MVK produced from photooxidation of other biogenic and alkene VOCs?

12. Section 2.5: This section is very confusing. For instance (not meant to be exhaustive), why is the OH calculated using equation 11 when Section 2.4 did the same thing? Where are the OH production rates calculated in equations 8-9 used? Why are the OH values then 'set' on lines 247-248?

13. Line 257: Avoid 'useless'.

14. Figure 2: Do the earlier measurements use the same/similar instrument as used here?

15. Line 296: 'slighter'?

16. Line 299-301: What is the point of this statement?

17. Line 314: How is 'haze' defined? Is there a textbook definition that is being used?

18. Lines 318-322: Unclear what this sentence means? 'front zone is in accordance with the increasing phase'? 'decreasing phase of pollution'? 'pressure pattern can be

clearly observed and used to explain the accumulation process'? All of these phrases make it extremely challenging to understand what is on the authors minds.

19. Line 334: What is the 'diurnal variation law'?

20. Line 351-353: 'the temperature influence on the evaporation rate can offset the daytime valley'. Not sure what is implied here.

21. Lines 360-362: Why aren't single-ring aromatics linked to a traffic source?

22. Line 368: Why was traffic restricted during the sampling time?

23. Line 404: What is the 'real situation'?

24. Line 407: Is the VOC/NOx ratio limited to the VOCs that were measured by the PTR-MS? If yes, say so.

25. Line 430: How can POA be oxidized in the gas-phase when POA is in the particle phase? Do you mean the semi-volatile vapors in equilibrium with POA?

26. Lines 484: Do you expect secondary production to be high with OH concentrations being so low (<10^6 molecules cm^-3)?

27. Lines 500 onwards: Were attempts made to explain the OOA (or SV-OOA) mass concentrations based on the SOA precursors measured by the PTR-MS? Are other SOA precursors (e.g., intermediately volatility organic compounds) relevant here?

---

## Author Comment (AC1) · 23 Jul 2020

Response to Anonymous Referee #1,

**Anonymous Referee #1**

The authors describe result of a field campaign in a city in Central China, in which VOCs were measured by online PTR-MS and by GC from canister samples. Additional information about meteorological parameters and inorganic compounds is given. The aim of the paper was to characterize sources of VOCs and their role for SOA formation. The design of the study only allowed observing a rather incomplete picture concerning the goal of the study. The limitations of the study and consequences for the results, however, are insufficiently discussed in the paper. The authors try to calculate various quantities such as chemical age of air masses, OH concentrations at the location of measurements, radical production rates using approaches that cannot be easily justified to be applicable for conditions of the campaign and with the limited number of measurements that were done. Overall, the paper reads as if the authors mainly try to calculate various parameters from measured values in a similar way as reported in literature, but without discussing the science behind. The main part of the paper is describing VOCs measurements, but does not give any new results about chemical processes or state of the atmosphere. VOC data are further analysis with respect to source of VOCs by a PMF analysis, which can be done with the limited number of measurements. Overall, however, the content of the paper is mainly a description of the level of a limited number of VOC concentration that were measured for one month in a city in China and attributing these VOCs to sources that are to be expected. Therefore, the paper does not contain sufficient new results to be published in ACP. In addition, scientific quality of the manuscript is not good enough, because limitations of applied methods are not discussed.

Thank you for your review. This paper is submitted as "measurement report" because of its limited scientific quality. Despite that, we think this study is worthwhile because it presents the VOCs characteristics and sources in the central China, where is within North China Plain and experiences heavily pollution in heating season. Up to now, the online observation data of VOCs, especially OVOCs are lack in this region, and this study would also provide data support for inventory verification and other scientific researches. Besides, SOA formation potential and radical generation capacity are calculated and firstly assigned to the resolved sources, which could propose the probable roles of benzenoids and OVOCs, and estimate the potential impacts from each source. Based on the calculation, we also find that VOCs take significant participant in SOA formation as precursors in photochemical reactions during clear and slight polluted days, while their roles in the enhancement of atmospheric oxidation capacity are more critical in heavily polluted days. In the manuscript, 38 VOC species are included, which are enough to reveal their roles in SOA formation in the investigated season, because benzenoids are recognized as the largest SOA contributors as precursors, and short-chain carbonyls have been found as most active VOC species for radical cycling. The investigated city, Xinxiang, is among the most polluted city in China. Although the description of VOCs concentration and source apportionment are based on one-month observation data, this study can expose the critical VOCs

emissions for SOA formation because the campaign was conducted in the intensive polluted period, which would be helpful for pollution mitigation in China. As for your questions and suggestions about the methods, the responses and the improvement plan are described below in details.

Some specific comments:

Radical sources: The authors claim that photolysis of OVOCs is the major source of radicals. This has been shown for a specific environment in the center of oil- gas-fields in the US, but cannot be generalized. The limited number of measurements in the campaign described in this paper does not allow to analyse source of radicals, neither appropriate radiation measurements were done nor important radical precursors such as nitrous acid were detected. Though mentioned in the text, the authors continue stating that carbonyl photolysis yields OH (Eq 9), which is not true.

In China, photolysis of OVOCs is a significant source of radicals and can determines the atmospheric oxidation capacity, and this conclusion has been reported in some recent studies (Liu et al., 2012; Xue et al., 2016; Yang et al., 2017). In this study, 610 hourly averaged samples with 44 species are involved in PMF model, which meets the requirements of source apportionment, and the potential radical contribution can be further estimated. Besides, the one-month measurement was conducted in the intensive polluted period, which can reveal the VOCs performance in pollution process in autumn and wintertime. As OH have been recognized mainly originating from the photolysis of $O_3$, OVOCs, and HONO (Crutzen et al., 1991), this study estimates and compares the photolysis and radical producing rates of the $O_3$ and carbonyls to reveal the significance of OVOCs in radical chemistry. Because of the limited measurement condition, the performance of nitric acid cannot be investigated currently, which will lead to some incompletion. Despite that, we think this study can still reveal the important roles of the OVOCs in the enhancement of atmospheric oxidation capacity in the ambient air, because the object of reference, $O_3$ photolysis, is among the biggest OH contributors in the studies in China (Yang et al., 2017; Feng et al., 2019). Eq. (9) have been applied in Beijing and Shenzhen in China, assuming that all the produced peroxy ($HO_2$ and $RO_2$) are completely converted into OH, which is the case during most of the daytime. (Rao et al., 2016; Wang et al., 2017). According to the comment, the existing conclusions about the significance of OVOC photolysis in China will be added, and the reason for the comparation between the photolysis rates of the $O_3$ and carbonyls, the limitation from the absence of nitric acid measurement, as well as the main point of the one-month measurement in autumn is described in the revised manuscript in line 104, 245-255, and 281-284.

OH concentration: The authors use a parameterization for calculating OH concentrations, for which important parameters like photolysis rates are only estimated. However, even if they had measurements, this approach is not applicable, because the parameterization was achieved for a totally different environment and location. There cannot be expectation that this parameterization can be used to calculate OH for the conditions of this campaign.

This parameterization has been applied for OH concentration calculation in Beijing and Shandong Province, and it is suggested valid in the previous studies (Yuan

et al., 2013; Zheng et al., 2011; Li et al., 2018). Besides, the previous study in Beijing estimated the uncertainty within in 48% in consideration of all factors. Because of the limitation of measurement, we could only use the Tropospheric Ultraviolet and Visible (TUV, version 5.0) model for photolysis frequencies estimation, following a study in Beijing, with rationality of parameter selection evaluated by sensitivity analysis (Rao et al., 2016). In order to minimize the inaccuracy in photolysis frequencies estimation, the AOD value will be recalculated using the model suggested in Beijing (Liang et al., 2013; Lin et al., 2013). The aim of this part is to compare the atmospheric oxidation capacity in aspect of OH between clear and hazy days, and the limitation, as well as uncertainty obtained from similar campaign are discussed in the revised manuscript (Section 2.5).

Chemical age: The authors use the ratio of isoprene to MVK+MACR to estimate the OH dose that give the chemical age of the air mass. This approach would only be applicable, if the air mass contained VOCs from approximately co-located emission sources that are transported to the measurement site without significant mixing of air masses from other sources. As the authors show there is little biogenic sources, but the majority of VOCs stem from anthropogenic sources. Therefore, it cannot be expected that the chemical age of anthropogenic sources can be estimated by using the degradation of isoprene which is emitted by plants. The location of the measurements is in a city, where anthropogenic sources of VOCs are presumably very close to the site. As a consequence, also the estimate for SOA production is based on questionable assumptions.

Thank you for your reminder. We will be estimated from the observed ratio between toluene and benzene, based on the assumption and description in the previous studies (de Gouw et al., 2005).

$$[OH]\Delta t = \frac{1}{k_{toluene} - k_{benzene}} \left[ ln\left( \left. \frac{[toluene]}{[benzene]} \right|_{t=0} \right) - ln\left( \left( \frac{[toluene]}{[benzene]} \right) \right) \right]$$

In this study, the ratio between toluene and benzene is lowest from 2:00 am to 4:00 am according to the daily change analysis, and the (toluene/benzene)$_{t=0}$ is estimated from the 95th percentile of the observed ratios during this period. Afterwards, the SOA production is re-estimated.

VOC measurements: The authors do not discuss consequences of the limited number of VOC species that were detected. For example, small alkenes, which often make a larger contribution to the total number of reactants in anthropogenic environments like here, are missing. Concerning the data quality there is only good agreement between PTR and GC measurements for benzene stated, but nothing said, if good agreement was also found for other species. Figure 2 gives a comparison of VOCs concentrations with other locations, but no conclusions can be drawn, because VOC levels highly depend on the distance to sources, time of the year etc. The authors do not make an attempt to give any interpretation, when they compare their VOC levels with those found in other locations. Figure 5 gives a correlation between observed TVOC and predicted TVOCs. It is insufficiently explained how predictions of TVOCs were derived, but it seems as if predictions rely on the PMF analysis. If this was the case, a

good correlation is certainly expected, because the PMF factors themselves are based on measured VOCs. The effect of prediction would only be smoothing out some of the variability of concentrations of single VOCs.

In this work, we can't get the information of some VOCs species including small alkanes and alkenes. However, although the small alkenes often make a large contribution in the ambient air, the missing of the measurement won't lead to great influences on SOA formation or radical generation estimations, because their yields for SOA formation are zero (Wu and Xie, 2018; Grosjean and Seinfeld, 1989), and there is no radical directly emitted from photolysis process. Similarly, the absence of small alkanes won't influence the further analysis. However, the missing of measurements of long-chain alkanes will lead to underestimation in SOAFP calculation, which have been discussed in the manuscript. The discussion about the missing of small alkenes will be added in the manuscript (line310-313,473-483).

The data quality can be achieved for both PTR and GC measurements because we calibrated the both two methods using standard gas cylinders. The species that can be detected from the both methods are rare, therefore the observed data from PTR-MS and GC measurement are incomparable in most cases. The concordance of the observed data from the two measurements is checked in order to make sure the combination of the two methods is feasible. We used benzene for consistency test because this species had considerable concentration in the ambient air, which could be effectively detected for both PTR and GC measurement. Besides, there is no isomers at the mass peaks of benzene in PTR mass spectra. The quality assurance for VOC measurement and the reason for choosing benzene for consistency test will be explained in details in the manuscript (Text S2, line 291-298).

According to your suggestions, we will give interpretations about the VOC concentration levels in line 331-352. Formaldehyde is different from the other species as its concentration is below the most observation data in the recent years. As the previous studies described, anthropogenic emission is the largest contributor of formaldehyde in wintertime, and the investigations in north China have shown that vehicle emission is the largest source in winter (Qian et al., 2019; Sheng et al., 2018). The source apportionment in the current study indicates that the vehicle exhaust is not intensive in Xinxiang which might because of the motor vehicle restrictions, and this would be the reason for the low concentration. Acetaldehyde and acetone are greatly influenced by industrial emissions in the current study, which is because that Xinxiang is heavily industrialized and this phenomenon is meeting the findings in other urban aeras (Singh et al., 1994; Yang et al., 2019b). Besides, some pharmaceutical factories are within a few kilometers from the observation site. Therefore, the above two species show a relatively high concentration compared with many locations. Similarly, benzenoids are abundantly emitted from industrial emissions, solvent evaporation, and residential heating in Xinxiang according to the source apportionment and diurnal variation, thereby the concentrations are also relatively high as well compared with other sites.

The comparison between observed TVOC and predicted TVOCs have been used in a few studies to get an overview for the result test of PMF model (Zheng et al., 2018;

Sarkar et al., 2017). In Figure 5 (shown as Fig S5 in the manuscript), TVOCs is calculated by summing up all the investigated VOC species, and the predicted VOCs are derived from PMF model. Although the predicted VOCs are estimated based on observed data, nonideal correlation will also occur if the model is not well run. According to the comments, the $R^2$ of all the species input in PMF model have been listed in Table S3, with a range of 0.41 to 0.97. Besides, the results of Base Model Displacement (DISP) and Bootstrap (BS) Error Estimation are demonstrated in Text S6 to reveal the effect of the prediction. Based on the above testing, the model result is proved reliable.

Technically: The quality of figures is poor due to small font sizes, small sizes of bars in bar plots and small legend sizes.

The figures have been adjusted to favorable sizes and layouts according to the suggestions.

Anonymous Referee #2

Zhang et al. performed online and offline VOC measurements using a PTR-MS and canister samples, respectively, in a central city in China. They compare their measurements to historical data, perform a factor analysis to identify primary VOC sources, and comment on the potential of the VOCs to form SOA. Based on a preliminary analysis, they find BTEX and acetone concentrations to be elevated compared to measurements elsewhere, solvent evaporation to be an important source for the measured VOCs, and benzenoid species to be important for SOA formation in the urban environment. There is a clear need for detailed online VOC measurements, on which analysis similar to those performed in this work would lead to an understanding of the primary sources to air pollution in populated cities. Hence the work is well motivated and the measurements are likely to be valuable to the atmospheric community. However, the analyses performed on the data are insufficiently described in terms of the methods and results. I generally recommend publication of this manuscript after the authors have had a chance to review my broad comments and responded to it.

Thank you for your review. According to your comments, the details of the methods and results of the measured data will be added in the revised manuscript, and the responses are presented as below.

Overall comments:

1. Measurement report: This article is submitted as a 'measurement report' but it does not seem to follow the data policy outlined on the ACP website: 'The data presented in measurement reports must be openly accessible in accordance with the EGU data policy.'. The data availability section in this paper currently reads 'The observational data in this study are available from the authors upon reasonable request (zhanghaixu@tsinghua.edu.cn).'. Please make sure that the data are available before final submission.

    The data will be uploaded according to the policy.

2. Data analysis: My understanding is that the measurement reports 'may include model results and conclusions of more limited scope than in research articles'. The authors, however, have begun to undertake a modest amount of analysis on the VOC measurements to study the source contributions, regional dependence, and implications for SOA. However, I do not think the analysis methods are presented clearly enough and/or the analysis results are described in detail for me to consider them complete; more detailed comments below. My interpretation of 'limited scope' in the context of this paper is to perform a 'complete' analysis for one of the tasks (e.g., source apportionment based on a PMF analysis) rather than a 'light' analysis for many tasks. In case this article is published with its current structure, it should be made clear that the analyses presented in this work are preliminary and will be continued to be worked on before they are published in their final form as a companion research article.

    The PMF model were conducted following the specification as a typical research article. The current version of the manuscript and supplementary material

show the variation of Q value as factor number set from 2 to 10, the details of factor profile, and the diurnal variation of each factor of the final result. According to the comment, the $r^2$ coefficients between observed values and predicted values of all the species will be added in the revised supplement (Table S3), and the results of two error estimation methods including Base Model Displacement (DISP) and Bootstrap (BS) Error Estimation have will be added to evaluate stability and uncertainty of the base run solution (Text S6). Besides, the characteristics and their potential contribution to SOA formation of each source, as well as the potential influential factors around, will be discussed in more details (line 212-271, Text S7). The source apportionment of OA is conducted preliminarily in the current study, because the only the fraction of OOA is needed when estimating the weights of VOCs in SOA formation as precursors. According to the comments, the above explanation has been added in the manuscript (line 179-182)

3. PTR-QMS: I know of a few earlier studies that have used a quadrupole PTR-MS to measure VOCs in urban environments (e.g., Warneke et al. (2012)). As the VOC species are measured at a unit mass resolution in a PTR-QMS, are the species identified and quantified in this work free from interferences? Depending on the answer to the interference question, can the data be used to perform PMF or would the interferences confound the source apportionment? For the reader's sake, can the authors cite earlier work that have used the PTR-MS measurements in the same manner as they are used here and state any limitations in performing such analysis? In addition, I am not sure how secondary production, in general, is resolved in PMF when both primary and secondary species are present in the same factor and when species within a factor have different reactivities.

   The quadrupole PTR-MS has been used in a few field studies (Li et al., 2019; Kaltsonoudis et al., 2016; Baudic et al., 2016). As previous comparison studies show that the most investigated species, such as methanol, formaldehyde, acetaldehyde, benzene, and toluene are influenced little by other species in the atmosphere, their interferences are usually neglected. The interference of isoprene has been discussed in previous studies, because furan from combustion of fossil fuels and waste may be higher in some urban aera (Li et al., 2019; de Gouw and Warneke, 2007). Besides, the interference of acetonitrile was also analyzed in some studies, but the interferences cannot be estimated (Baudic et al., 2016; de Gouw et al., 2003; Dunne et al., 2012). In the revised manuscript, interferences of VOCs used for radical generation, SOA formation calculation, and as critical tracers in source apportionment are discussed based on the experience in the previous studies, in Text S1.

   On the whole, most of the mass peaks investigated in the current study are dominated by the targeted species. Therefore, the result of source apportionment won't be influenced greatly. On the other side, the interferences can be separated and observed in the PMF result. For example, acetonitrile is abundantly contributed by industrial emission and residential heating in the current study, indicating that except for emitted from coal and biomass combustion, interferences from alkanes (C3−C5) emitted from industrial sources are significant as well. For m/z 69, the

signal intensity is significantly attributed to solvent evaporation, industrial emission, and thermal power generation, which might own to several reasons. First, isoprene is probably emitted from anthropogenic sources, such as vehicle exhaust and synthetic rubber industry (Gil'manov et al., 2010; Barletta et al., 2002). Then, anthropogenic furan and cycloalkanes from biomass burning plumes, combustion of fossil fuels and waste, as well as oil/gas evaporation might contribute negligible fractions of the signal of m/z 69. The m/z 71 is recognized as oxidation products of isoprene, and the industrial emission, solvent evaporation, and thermal power generation have been found the largest contribution according to the PMF model, which is in accordance with m/z 69. Therefore, secondary products are mixed into the resolved anthropogenic source in the current study. Besides, the oxygenated VOCs (C>3) are usually considered from photochemical process, but they are abundantly attributed to industrial emission, solvent evaporation, and thermal power generation in the PMF result. It should be explained that although the different precursors have different reactivities, their consumption and the products formation rates have positive correlation to the concentrations of oxidants, temperature, and luminous intensity. The PMF model resolve the factors based on the mathematical patterns of the signals, and the sources with similar variation characteristics are hard to separated. Therefore, the secondary products are likely resolved into the sources which have similar variation regular. Similar results can be found in a few previous studies (Guha et al., 2015; Bon et al., 2011; Stojić et al., 2015; Yang et al., 2019a). The m/z 107 is mainly contributed by C8 aromatics, which are abundantly evaporate from solvent usage, while the interferences from benzaldehyde which has been reported more important in aged air masses can't be estimated because secondary sources are not separated (Warneke et al., 2003). As the campaign is conducted in late autumn and winter, the VOCs are currently assigned to primary species. Above all, industrial emission, solvent evaporation, and thermal power generation are speculated more likely mixed with secondary products. Therefore, the fraction of some species such as C8 aromatics for the three sources might be overestimated. In the revised manuscript, the influence on the PMF result from interferences of VOCs are analyzed in line 141-147 and Text S9.

Confusing phrases: Throughout the manuscript, there are lots of instances where it was very hard to understand what was being said. The technical communication needs to be substantially improved in the revision.

According to the comments, the confusing phrases have been revised, and the detailed responses can be referred below.

Detailed technical comments:

1. Line 23: what does 'obviously' mean?

It means that all the observed signals are higher than the detection limit, in the revised manuscript, the confusing phrase is deleted.

2. Line 26: what does 'high levels' mean? A recurring comment is that the authors describe trends qualitatively, which doesn't help with the interpretation.

In the abstract, we want to describe that the concentration of the BTEX, acetaldehyde, and acetone are relatively high compared with other studies. In the

revised manuscript, the percentiles of the measured data among the previous studies are showed in line 25-27. Accordingly, the concentrations of BTEX, acetaldehyde, and acetone are higher than 50 percentiles of previous data, which means that the anthropogenic emissions in Xinxiang are intensive.

3. Line 28: Are benzenoids only reduced single-ring aromatics in this work? If yes, please specify that or use 'single-ring aromatics'.

The benzenoids in our study are mainly single-ring aromatics, which contribute SOA formation potential most according to previous studies. Besides, naphthalene is included in PMF model. In the revised manuscript, the term "benzenoids" is specified in line 29 and 171.

4. Line 58: '20-30%' of what? Another recurring comment is the text does not provide context for what is increasing or decreasing and the reader is left guessing (sometimes incorrectly).

It means that the total concentration of aromatic hydrocarbons constitutes 20-30% of the total identified VOC concentration. In the revised manuscript this description and the expression about what is increasing and decreasing in the full text will be described in more details.

5. Lines 68-71: That the 2-product or n-product (i.e., volatility basis set) SOA model has not been used to interpret field data is misleading. There is extensive literature over the past two decades that points to the use of 2/n-product models for field analysis (e.g.,Dzepina et al., 2009).

Thank you for your reminder, and the expression has been corrected in the revised manuscript.

6. Line 81: What are these 'differences in different regions'? Are there some universal trends in ranking of the sources?

Vehicle exhaust emissions, industry emissions, fossil fuel volatilization, the use of chemical reagents, and biomass combustion are important sources of atmospheric VOCs, but the fractions of each sources vary in different regions because of the industrial structure and power consumption. In general, vehicle exhaust and solvent usage are the top sources of VOCs in the areas of dense population and megacities (Yuan et al., 2013; Fu et al., 2016; Gao et al., 2019). But in some specific regions and seasons, the dominant sources are different. For example, in an oil- and gas-bearing basin, the main VOC source is natural gas (Zheng et al., 2018), and in wintertime in Beijing, coal combustion is among the top sources of VOC emission (Liu et al., 2017). The detailed explanation has been added in the manuscript in line 85-94.

7. Line 93: What is 'autumn'? Why not just specify the dates?

It has been revised in the manuscript in line 104-105.

8. Line 123: 'as focus or tracers are'? Please correct.

In the revised manuscript, this expression has been replaced as "the VOCs used for radical generation, SOA formation calculation, and as critical tracers in source apportionment" in the revised manuscript in lines 135-136.

9. Sections 2.2-2.3: My general sense is that the detailed provided in these sections is not sufficient, especially if the reader is not familiar with the techniques.

The details of the principle and conduction of PMF model, as well as PSCF and

CWT calculations based on trajectory analysis have been described in the supplement. In the revised version, more details of the operation are added in lines 211-216 and Text S7.

10. Line 187-188: My sense is that PSCF and CWT need to be redefined in the text, separate from the abstract.

The definition of PSCF (potential source contribution function) and CWT (concentration weighted trajectory) has been mentioned in the revised text in line 213.

11. Lines 210-220: The other technique I have seen is the toluene/benzene ratio. Was there a reason why isoprene, MACR, and MVK were used to deduce OH exposures? Also, are MACR and MVK produced from photooxidation of other biogenic and alkene VOCs?

In the previous version, we used isoprene, MACR, and MVK to deduce OH exposures in order to minimize the impacts of anthropogenic emissions. However, this method has many defects in the urban site in winter according to the two referees' comments and the application condition mentioned in previous studies. Therefore, OH exposures have been recalculated using toluene/benzene ratio in the revised manuscript.

12. Section 2.5: This section is very confusing. For instance (not meant to be exhaustive), why is the OH calculated using equation 11 when Section 2.4 did the same thing? Where are the OH production rates calculated in equations 8-9 used? Why are the OH values then 'set' on lines 247-248?

This section aims to describe the method for radical contribution calculation, and the main formulas are eq. (8)-(10). In Eq. (10), the concentration of OH is required, hence OH is calculated using Eq. (11). This equation is different from the section 2.4, because the OH exposures in the previous section represents the degree of photooxidation of VOC. In order to compare the radical production capacities of the investigated OVOCs in clear and polluted periods, the average OH values in clear and polluted days were set. In the revised manuscript, real-time OH values are used, and the average radical production rates between clear and polluted periods of each species are compared. In order to eliminate the confusion, this section is rearranged.

According to the comments, this word has been revised as "not effective" in the manuscript in line 288.

14. Figure 2: Do the earlier measurements use the same/similar instrument as used here?

The earlier measurements referred in the Fig.2 are mostly obtained from PTR-MS, the details and the discussion of the comparison have been added in the revised manuscript in lines 321-326.

In this study, the enhancement ratios of formaldehyde, toluene, and C8 aromatics from clear to hazy days are relatively low compared with acetaldehyde, acetone, and benzene. Correspondingly, the photolysis rate of formaldehyde, and the photochemical rates of toluene and C8 aromatics are higher than the other species. Therefore, we speculate that photolysis and photochemical might result in the differences of the enhancement among the different species. This statement has been added in the revised manuscript in lines 348-353.

16. Line 299-301: What is the point of this statement?

In this section, influences of meteorological conditions on the VOCs concentrations are discussed. Generally speaking, pollutant dispersion is primarily related to wind direction and speed, while atmospheric chemical reactions can be influenced by temperature and humidity. In the following text, the potential reasons for the VOCs concentrations variation are discussed based on the known rules and the observed meteorological factors. The point of this statement has been clarified in the revised manuscript in lines 355-358.

17. Line 314: How is 'haze' defined? Is there a textbook definition that is being used?

Strictly speaking, haze is defined as the phenomenon when horizontal visibility is less than 10 km, with abundant particles suspended in the atmosphere (Wu, 2011). In the current study, the data of horizontal visibility is not obtained, and the hazy days mainly point to the polluted period. In order to minimize the confusion, the expressions of "haze" in the revised manuscript have been changed into "polluted".

18. Lines 318-322: Unclear what this sentence means? 'front zone is in accordance with the increasing phase'? 'decreasing phase of pollution'? 'pressure pattern can be clearly observed and used to explain the accumulation process'? All of these phrases make it extremely challenging to understand what is on the authors minds.

The first two phrases are quoted from a previous study on the relationship between atmospheric pollution processes and synoptic pressure patterns in northern China (Chen et al., 2008). Based on the studies in 10 cities in northern China, a regular pattern has been found that the air pollutants are generally accumulated in the phase of high pressure, and stored in the phase of low pressure to reach maximum values, and then the pollutants will be dispersed when front came, completing a cycle of air pollution process. In the current study, the pressure patterns with high pressure and the successive low pressure occur on 7-9, 16-20, and 24-26 November, and the pollutants' accumulation process are observed simultaneously. Therefore, the meteorological impacts can be clearly explained. In the revised manuscript, the detailed description has been added in lines 375-384.

19. Line 334: What is the 'diurnal variation law'?

The temperature shows regular diurnal variation, resulting in reduced atmospheric diffusion in the morning and best diffusion in the afternoon. Therefore, the concentrations of pollutants tend to be accumulated in the morning and dispersed in the afternoon. Besides, the tracers from particular sources shows distinct diurnal characteristics. For example, $NO_X$ and CO show the bimodal features of vehicle emissions. This statement has been revised to avoid confusion in the manuscript in lines 396-399 and Text S8.

20. Line 351-353: 'the temperature influence on the evaporation rate can offset the daytime valley'. Not sure what is implied here.

As it described in Section 3.2, the concentrations of pollutants tend to be accumulated in the morning and dispersed in the afternoon owning to the variation of atmospheric diffusion. Therefore, there should be a "concentration valley" in the afternoon if the emission rates of the VOCs are steady. However, VOCs evaporation is highly associated with temperature, and the abundant emissions in the afternoon offset

the daytime valley. The explanation has been added in the revised manuscript in lines 409-414.

21. Lines 360-362: Why aren't single-ring aromatics linked to a traffic source?

Thank you for your reminder, and the anthropogenic emissions of single-ring aromatics have been added in the manuscript in line 419.

22. Line 368: Why was traffic restricted during the sampling time?

It is a government decision in order to mitigate pollutant concentration during the periods of frequent pollution, and the explanation has been added in the manuscript in lines 426-428.

23. Line 404: What is the 'real situation'?

It is that many plants are located in the remote districts around the urban area. In the manuscript, the phrase "actual situation" may be more suitable and easier to understand, and it has been revised in lines 458-460.

24. Line 407: Is the VOC/NOx ratio limited to the VOCs that were measured by the PTR-MS? If yes, say so.

Yes. The VOCs were measure by the PTR-MS and selected as their SOA yields are greater than zero. The explanation has been added in the manuscript in line 465.

25. Line 430: How can POA be oxidized in the gas-phase when POA is in the particle phase? Do you mean the semi-volatile vapors in equilibrium with POA?

It has been reported that some species of POA can evaporate to the atmosphere and can be oxidized further, and then the oxidation products will re-partition into aerosols (Xing et al., 2019). The detailed description has been added in the manuscript in line 490-491.

26. Lines 484: Do you expect secondary production to be high with OH concentrations being so low (<10ˆ6 molecules cmˆ-3)?

Some previous studies in in Birmingham, NewYork, Tokyo, and Beijing have shown that the OH concentrations are generally within the range of $1.2–2.0 \times 10^6$ cm$^{-3}$ (Ma et al., 2019). Besides, the OH concentrations are usually found dropping to $0.3\times 10^6$ cm$^{-3}$ in polluted period in the northern China (Rao et al., 2016; Lu et al., 2013; Zheng et al., 2011). Comparatively, the OH concentration in the clear period is low, while the OH concentration in the polluted period is pretty close to the similar campaign. Previous field studies have estimated that the contribution of toluene in SOA ranges from 15% to 79% in megacities in China, by utilizing tracer-based method (Gao et al., 2019). In the current study, the average estimated secondary production of toluene is 35%, which is within the normal range. The comparation and discussion will be added in the manuscript in lines 507-512, lines 499-501.

27. Lines 500 onwards: Were attempts made to explain the OOA (or SV-OOA) mass concentrations based on the SOA precursors measured by the PTR-MS? Are other SOA precursors (e.g., intermediately volatility organic compounds) relevant here?

In the current study, the contribution of OOA mass concentrations based on the SOA precursors are investigated including benzene and toluene because aromatics are reported as the biggest SOA contributor among VOCs, and the above two species have the biggest SOAFP contributions in the current study (in lines 59-61, 501-503). Intermediately volatility organic compounds (IVOCs) may be important SOA

precursors, but they cannot be measured in the current study, and this limitation has been mentioned in the manuscript in line 480-483.

**References:**

Barletta, B., Meinardi, S., Simpson, I. J., Khwaja, H. A., Blake, D. R., and Rowland, F. S.: Mixing ratios of volatile organic compounds (VOCs) in the atmosphere of Karachi, Pakistan, Atmospheric Environment, 36, 3429-3443, https://doi.org/10.1016/S1352-2310(02)00302-3, 2002.

Baudic, A., Gros, V., Sauvage, S., Locoge, N., Sanchez, O., Sarda-Estève, R., Kalogridis, C., Petit, J. E., Bonnaire, N., Baisnée, D., Favez, O., Albinet, A., Sciare, J., and Bonsang, B.: Seasonal variability and source apportionment of volatile organic compounds (VOCs) in the Paris megacity (France), Atmos. Chem. Phys., 16, 11961-11989, 10.5194/acp-16-11961-2016, 2016.

Bon, D. M., Ulbrich, I. M., de Gouw, J. A., Warneke, C., Kuster, W. C., Alexander, M. L., Baker, A., Beyersdorf, A. J., Blake, D., Fall, R., Jimenez, J. L., Herndon, S. C., Huey, L. G., Knighton, W. B., Ortega, J., Springston, S., and Vargas, O.: Measurements of volatile organic compounds at a suburban ground site (T1) in Mexico City during the MILAGRO 2006 campaign: measurement comparison, emission ratios, and source attribution, Atmos. Chem. Phys., 11, 2399-2421, 10.5194/acp-11-2399-2011, 2011.

Chen, Z. H., Cheng, S. Y., Li, J. B., Guo, X. R., Wang, W. H., and Chen, D. S.: Relationship between atmospheric pollution processes and synoptic pressure patterns in northern China, Atmospheric Environment, 42, 6078-6087, https://doi.org/10.1016/j.atmosenv.2008.03.043, 2008.

Crutzen, P. J., Zimmermann, P. H. J. T. S. B.-c., and Meteorology, P.: The changing photochemistry of the troposphere, 43, 136-151, 1991.

de Gouw, J., and Warneke, C.: Measurements of volatile organic compounds in the earth's atmosphere using proton-transfer-reaction mass spectrometry, 26, 223-257, 10.1002/mas.20119, 2007.

de Gouw, J. A., Warneke, C., Parrish, D. D., Holloway, J. S., Trainer, M., and Fehsenfeld, F. C.: Emission sources and ocean uptake of acetonitrile (CH3CN) in the atmosphere, 108, 10.1029/2002jd002897, 2003.

de Gouw, J. A., Middlebrook, A. M., Warneke, C., Goldan, P. D., Kuster, W. C., Roberts, J. M., Fehsenfeld, F. C., Worsnop, D. R., Canagaratna, M. R., Pszenny, A. A. P., Keene, W. C., Marchewka, M., Bertman, S. B., and Bates, T. S.: Budget of organic carbon in a polluted atmosphere: Results from the New England Air Quality Study in 2002, Journal of Geophysical Research-Atmospheres, 110, 10.1029/2004jd005623, 2005.

Ding, X., He, Q.-F., Shen, R.-Q., Yu, Q.-Q., and Wang, X.-M.: Spatial distributions of secondary organic aerosols from isoprene, monoterpenes, beta-caryophyllene, and aromatics over China during summer, Journal of Geophysical Research-Atmospheres, 119, 11877-11891, 10.1002/2014jd021748, 2014.

Dunne, E., Galbally, I. E., Lawson, S., and Patti, A.: Interference in the PTR-MS measurement of acetonitrile at m/z 42 in polluted urban air—A study using switchable reagent ion PTR-MS, International Journal of Mass Spectrometry, 319-320, 40-47, https://doi.org/10.1016/j.ijms.2012.05.004, 2012.

Dzepina, K., Volkamer, R. M., Madronich, S., Tulet, P., Ulbrich, I. M., Zhang, Q.,

Cappa, C. D., Ziemann, P. J., and Jimenez, J. L.: Evaluation of recently-proposed secondary organic aerosol models for a case study in Mexico City, Atmospheric Chemistry and Physics, 9, 5681-5709, 10.5194/acp-9-5681-2009, 2009.

Feng, T., Zhao, S., Bei, N., Wu, J., Liu, S., Li, X., Liu, L., Qian, Y., Yang, Q., Wang, Y., Zhou, W., Cao, J., and Li, G.: Secondary organic aerosol enhanced by increasing atmospheric oxidizing capacity in Beijing–Tianjin–Hebei (BTH), China, Atmos. Chem. Phys., 19, 7429-7443, 10.5194/acp-19-7429-2019, 2019.

Fu, P., Aggarwal, S. G., Chen, J., Li, J., Sun, Y., Wang, Z., Chen, H., Liao, H., Ding, A., Umarji, G. S., Patil, R. S., Chen, Q., and Kawamura, K.: Molecular Markers of Secondary Organic Aerosol in Mumbai, India, Environmental Science & Technology, 50, 4659-4667, 10.1021/acs.est.6b00372, 2016.

Gao, Y., Wang, H., Zhang, X., Jing, S. a., Peng, Y., Qiao, L., Zhou, M., Huang, D. D., Wang, Q., Li, X., Li, L., Feng, J., Ma, Y., and Li, Y.: Estimating Secondary Organic Aerosol Production from Toluene Photochemistry in a Megacity of China, Environmental science & technology, 53, 8664-8671, 10.1021/acs.est.9b00651, 2019.

Gil'manov, K. K., Romanova, R. G., Lamberov, A. A., and Gil'mullin, R. R.: Isoprene manufacturing process on a new bimetallic (platinum-tin) catalyst, Petroleum Chemistry, 50, 388-394, 10.1134/s0965544110050129, 2010.

Grosjean, D., and Seinfeld, J. H.: Parameterization of the formation potential of secondary organic aerosols, Atmospheric Environment, 23, 1733-1747, 10.1016/0004-6981(89)90058-9, 1989.

Guha, A., Gentner, D. R., Weber, R. J., Provencal, R., and Goldstein, A. H.: Source apportionment of methane and nitrous oxide in California's San Joaquin Valley at CalNex 2010 via positive matrix factorization, Atmos. Chem. Phys., 15, 12043-12063, 10.5194/acp-15-12043-2015, 2015.

Kaltsonoudis, C., Kostenidou, E., Florou, K., Psichoudaki, M., and Pandis, S. N.: Temporal variability and sources of VOCs in urban areas of the eastern Mediterranean, Atmos. Chem. Phys., 16, 14825-14842, 10.5194/acp-16-14825-2016, 2016.

Li, D., Xue, L., Wen, L., Wang, X., Chen, T., Mellouki, A., Chen, J., and Wang, W.: Characteristics and sources of nitrous acid in an urban atmosphere of northern China: Results from 1-yr continuous observations, Atmospheric Environment, 182, 296-306, https://doi.org/10.1016/j.atmosenv.2018.03.033, 2018.

Li, K., Li, J., Tong, S., Wang, W., Huang, R.-J., and Ge, M.: Characteristics of wintertime VOCs in suburban and urban Beijing: concentrations, emission ratios, and festival effects, Atmospheric Chemistry and Physics, 19, 8021-8036, 10.5194/acp-19-8021-2019, 2019.

Liang, H., Chen, Z. M., Huang, D., Zhao, Y., and Li, Z. Y.: Impacts of aerosols on the chemistry of atmospheric trace gases: a case study of peroxides and $HO_2$ radicals, Atmos. Chem. Phys., 13, 11259-11276, 10.5194/acp-13-11259-2013, 2013.

Lin, H. F., Xin, J. Y., Zhang, W. Y., Wang, Y. S., and Chen, C. L. J. E. S.: Comparison of atmospheric particulate matter and aerosol optical depth in Beijing City, 2013.

Liu, C., Ma, Z., Mu, Y., Liu, J., Zhang, C., Zhang, Y., Liu, P., and Zhang, H.: The levels,

variation characteristics, and sources of atmospheric non-methane hydrocarbon compounds during wintertime in Beijing, China, Atmos. Chem. Phys., 17, 10633-10649, 10.5194/acp-17-10633-2017, 2017.

Liu, Z., Wang, Y., Gu, D., Zhao, C., Huey, L. G., Stickel, R., Liao, J., Shao, M., Zhu, T., Zeng, L., Amoroso, A., Costabile, F., Chang, C. C., and Liu, S. C.: Summertime photochemistry during CAREBeijing-2007: $RO_x$ budgets and $O_3$ formation, Atmos. Chem. Phys., 12, 7737-7752, 10.5194/acp-12-7737-2012, 2012.

Lu, K. D., Hofzumahaus, A., Holland, F., Bohn, B., Brauers, T., Fuchs, H., Hu, M., Häseler, R., Kita, K., Kondo, Y., Li, X., Lou, S. R., Oebel, A., Shao, M., Zeng, L. M., Wahner, A., Zhu, T., Zhang, Y. H., and Rohrer, F.: Missing OH source in a suburban environment near Beijing: observed and modelled OH and $HO_2$ concentrations in summer 2006, Atmos. Chem. Phys., 13, 1057-1080, 10.5194/acp-13-1057-2013, 2013.

Ma, X., Tan, Z., Lu, K., Yang, X., Liu, Y., Li, S., Li, X., Chen, S., Novelli, A., Cho, C., Zeng, L., Wahner, A., and Zhang, Y.: Winter photochemistry in Beijing: Observation and model simulation of OH and HO2 radicals at an urban site, Science of The Total Environment, 685, 85-95, https://doi.org/10.1016/j.scitotenv.2019.05.329, 2019.

Qian, X., Shen, H. Q., and Chen, Z. M.: Characterizing summer and winter carbonyl compounds in Beijing atmosphere, Atmospheric Environment, 214, 10, 10.1016/j.atmosenv.2019.116845, 2019.

Rao, Z., Chen, Z., Liang, H., Huang, L., and Huang, D.: Carbonyl compounds over urban Beijing: Concentrations on haze and non-haze days and effects on radical chemistry, Atmospheric Environment, 124, 207-216, 10.1016/j.atmosenv.2015.06.050, 2016.

Sarkar, C., Sinha, V., Sinha, B., Panday, A. K., Rupakheti, M., and Lawrence, M. G.: Source apportionment of NMVOCs in the Kathmandu Valley during the SusKat-ABC international field campaign using positive matrix factorization, Atmos. Chem. Phys., 17, 8129-8156, 10.5194/acp-17-8129-2017, 2017.

Sheng, J. J., Zhao, D. L., Ding, D. P., Li, X., Huang, M. Y., Gao, Y., Quan, J. N., and Zhang, Q.: Characterizing the level, photochemical reactivity, emission, and source contribution of the volatile organic compounds based on PTR-TOF-MS during winter haze period in Beijing, China, Atmospheric Research, 212, 54-63, 10.1016/j.atmosres.2018.05.005, 2018.

Singh, H. B., O'Hara, D., Herlth, D., Sachse, W., Blake, D. R., Bradshaw, J. D., Kanakidou, M., and Crutzen, P. J.: Acetone in the atmosphere: Distribution, sources, and sinks, 99, 1805-1819, 10.1029/93jd00764, 1994.

Stojić, A., Stanišić Stojić, S., Mijić, Z., Šoštarić, A., and Rajšić, S.: Spatio-temporal distribution of VOC emissions in urban area based on receptor modeling, Atmospheric Environment, 106, 71-79, https://doi.org/10.1016/j.atmosenv.2015.01.071, 2015.

Wang, C., Huang, X.-F., Han, Y., Zhu, B., and He, L.-Y.: Sources and Potential Photochemical Roles of Formaldehyde in an Urban Atmosphere in South China, 122, 11,934-911,947, 10.1002/2017jd027266, 2017.

Warneke, C., de Gouw, J. A., Kuster, W. C., Goldan, P. D., and Fall, R.: Validation of Atmospheric VOC Measurements by Proton-Transfer- Reaction Mass Spectrometry Using a Gas-Chromatographic Preseparation Method, Environmental Science & Technology, 37, 2494-2501, 10.1021/es026266i, 2003.

Wu, D.: Formation and Evolution of Haze Weather, Enuivonmental Science and Technology, 34, 157-161, 2011.

Wu, R., and Xie, S.: Spatial Distribution of Secondary Organic Aerosol Formation Potential in China Derived from Speciated Anthropogenic Volatile Organic Compound Emissions, Environmental Science & Technology, 52, 8146-8156, 10.1021/acs.est.8b01269, 2018.

Xing, L., Wu, J., Elser, M., Tong, S., Liu, S., Li, X., Liu, L., Cao, J., Zhou, J., El-Haddad, I., Huang, R., Ge, M., Tie, X., Prevot, A. S. H., and Li, G.: Wintertime secondary organic aerosol formation in Beijing-Tianjin-Hebei (BTH): contributions of HONO sources and heterogeneous reactions, Atmospheric Chemistry and Physics, 19, 2343-2359, 10.5194/acp-19-2343-2019, 2019.

Xue, L., Gu, R., Wang, T., Wang, X., Saunders, S., Blake, D., Louie, P. K. K., Luk, C. W. Y., Simpson, I., Xu, Z., Wang, Z., Gao, Y., Lee, S., Mellouki, A., and Wang, W.: Oxidative capacity and radical chemistry in the polluted atmosphere of Hong Kong and Pearl River Delta region: analysis of a severe photochemical smog episode, Atmos. Chem. Phys., 16, 9891-9903, 10.5194/acp-16-9891-2016, 2016.

Yang, X., Xue, L., Yao, L., Li, Q., Wen, L., Zhu, Y., Chen, T., Wang, X., Yang, L., Wang, T., Lee, S., Chen, J., and Wang, W.: Carbonyl compounds at Mount Tai in the North China Plain: Characteristics, sources, and effects on ozone formation, Atmospheric Research, 196, 53-61, 10.1016/j.atmosres.2017.06.005, 2017.

Yang, Y., Ji, D., Sun, J., Wang, Y., Yao, D., Zhao, S., Yu, X., Zeng, L., Zhang, R., Zhang, H., Wang, Y., and Wang, Y.: Ambient volatile organic compounds in a suburban site between Beijing and Tianjin: Concentration levels, source apportionment and health risk assessment, Science of The Total Environment, 695, 133889, https://doi.org/10.1016/j.scitotenv.2019.133889, 2019a.

Yang, Z., Cheng, H. R., Wang, Z. W., Peng, J., Zhu, J. X., Lyu, X. P., and Guo, H.: Chemical characteristics of atmospheric carbonyl compounds and source identification of formaldehyde in Wuhan, Central China, Atmospheric Research, 228, 95-106, 10.1016/j.atmosres.2019.05.020, 2019b.

Yuan, B., Hu, W. W., Shao, M., Wang, M., Chen, W. T., Lu, S. H., Zeng, L. M., and Hu, M.: VOC emissions, evolutions and contributions to SOA formation at a receptor site in eastern China, Atmospheric Chemistry and Physics, 13, 8815-8832, 10.5194/acp-13-8815-2013, 2013.

Zheng, H., Kong, S., Xing, X., Mao, Y., Hu, T., Ding, Y., Li, G., Liu, D., Li, S., and Qi, S.: Monitoring of volatile organic compounds (VOCs) from an oil and gas station in northwest China for 1 year, Atmos. Chem. Phys., 18, 4567-4595, 10.5194/acp-18-4567-2018, 2018.

Zheng, J., Hu, M., Zhang, R., Yue, D., Wang, Z., Guo, S., Li, X., Bohn, B., Shao, M., He, L., Huang, X., Wiedensohler, A., and Zhu, T.: Measurements of gaseous $H_2SO_4$ by AP-ID-CIMS during CAREBeijing 2008 Campaign, Atmos. Chem.

Phys., 11, 7755-7765, 10.5194/acp-11-7755-2011, 2011.